# Iron oxide nanozymes stabilize stannous fluoride for targeted biofilm killing and synergistic oral disease prevention

Yue Huang[1,2,3,15], Yuan Liu[2,4,15], Nil Kanatha Pandey[1,2,3,15], Shrey Shah[1,5], Aurea Simon-Soro[2,3,6], Jessica C. Hsu[1,5], Zhi Ren[2,3,7], Zhenting Xiang[2,3], Dongyeop Kim[2,3,8], Tatsuro Ito[2,3,9], Min Jun Oh[2,3,10], Christine Buckley[11], Faizan Alawi[12], Yong Li[2,3], Paul J. M. Smeets[13,14], Sarah Boyer[14], Xingchen Zhao[14], Derk Joester[14], Domenick T. Zero[11], David P. Cormode[1,5] ✉ & Hyun Koo[2,3,7] ✉

Dental caries is the most common human disease caused by oral biofilms despite the widespread use of fluoride as the primary anticaries agent. Recently, an FDA-approved iron oxide nanoparticle (ferumoxytol, Fer) has shown to kill and degrade caries-causing biofilms through catalytic activation of hydrogen peroxide. However, Fer cannot interfere with enamel acid demineralization. Here, we show notable synergy when Fer is combined with stannous fluoride ($SnF_2$), markedly inhibiting both biofilm accumulation and enamel damage more effectively than either alone. Unexpectedly, we discover that the stability of $SnF_2$ is enhanced when mixed with Fer in aqueous solutions while increasing catalytic activity of Fer without any additives. Notably, Fer in combination with $SnF_2$ is exceptionally effective in controlling dental caries in vivo, even at four times lower concentrations, without adverse effects on host tissues or oral microbiome. Our results reveal a potent therapeutic synergism using approved agents while providing facile $SnF_2$ stabilization, to prevent a widespread oral disease with reduced fluoride exposure.

Dental caries (tooth decay) is the most prevalent and costly biofilm-induced oral disease that causes the destruction of the mineralized tooth tissue[1]. In caries-inducing (cariogenic) biofilms, microorganisms form highly protected biostructures that create acidic pH micro-environments, promoting cariogenic bacteria growth and acid dissolution of tooth-enamel[2,3]. Despite increased use of fluoride (the mainstay anticaries agent), it remains unresolved and affects 3.1 billion people worldwide, with annual costs exceeding US $290 billion[4,5]. Even though fluoride is effective in reducing tooth enamel demineralization at acidic pH values[6,7], it has limited antibiofilm activity despite inhibitory effects against planktonic bacteria[8]. Additionally, current modalities, including high-dose fluoride treatments, are insufficient to prevent dental caries in high-risk individuals or individuals with a low salivary flow/buffering capacity where pathogenic dental biofilms rapidly accumulate under sugar-rich diets and poor oral hygiene that enables firm bacterial adhesion to teeth[9–11]. Although currently used fluoride doses provide minimal toxicity, there are potential risks associated with fluoride overexposure (e.g., dental fluorosis), especially for young children[12–16].

Ferumoxytol (Fer), an aqueous iron oxide nanoparticle formulation approved by the Food and Drug Administration (FDA) for systemic treatment of iron deficiency, has shown both efficacy and specificity against cariogenic biofilms when used topically, through selective pathogen binding and acidic pH-activation of hydrogen peroxide ($H_2O_2$) via catalytic (peroxidase-like) activity[17,18]. Although topical applications of Fer can reduce dental caries in vivo, it does not interfere physiochemically with enamel demineralization and is unable to entirely prevent the progression of the disease. To improve the

efficacy of Fer, combination with fluoride could potentiate the therapeutic effects. We hypothesized that Fer and fluoride could complement each other's properties, even at lower concentrations, to target the development of dental caries more effectively without increasing fluoride exposure.

Herein, we evaluated the combination of Fer with two formulations of fluoride widely used in oral health care, sodium fluoride (NaF) and stannous fluoride ($SnF_2$). While the combination with NaF did not show improvement, when Fer was combined with $SnF_2$ we observed remarkable synergistic effects in vitro and in vivo. We found that $SnF_2$ was highly stable in aqueous solution when mixed with Fer; the lack of stability of $SnF_2$ has been a major limitation in commercial formulations, requiring use of chemical additives[19,20]. Unexpectedly, the catalytic activity of Fer significantly increased when mixed with $SnF_2$, thereby enhancing antimicrobial potency. Further analysis revealed that $Sn^{2+}$ was bound by carboxylate groups in the carboxymethyl-dextran coating of Fer, thereby enhancing the stability of $SnF_2$. When tested in a rodent model, we found that Fer in combination with $SnF_2$ was remarkably effective in preventing dental caries (substantially superior to either alone), completely blocking enamel cavitation, an outcome not observed before. Moreover, the anticaries efficacy was achieved at four times lower dosage of $SnF_2$. Notably, fluoride, iron, and tin were detected on the outer layers of the enamel, indicating codelivery to form a caries-protective film in situ. Altogether, we developed a combination therapy with unexpected synergistic mechanisms that target the biological (biofilm) and physicochemical (enamel demineralization) traits of dental caries while providing a facile $SnF_2$ stabilization and lower dosage strategy against a widespread and costly oral disease, as summarized in Fig. 1.

## Results

### Antibiofilm activity of Fer in combination with $SnF_2$ in vitro

Fluoride is widely used as a gold standard anticaries agent, but it does not provide full protection, especially in severe cases where pathogenic biofilms rapidly accumulate. Despite its limited antibiofilm activity, sodium fluoride (NaF) can affect bacterial glycolysis and acid tolerance[21–23], whereas stannous fluoride ($SnF_2$) provides stronger antibacterial activity imparted by $Sn^{2+}$ ions[24,25]. First, we tested the antibiofilm activity of both NaF and $SnF_2$ (1000 ppm of F, the typical concentration in over-the-counter formulations) and found that $SnF_2$ can significantly inhibit *Streptococcus mutans* (*S. mutans*, a cariogenic pathogen) viability and reduce the biomass more effectively than NaF (Supplementary Fig. 1a, b). Afterward, we combined NaF or $SnF_2$ with Fer (1 mg of Fe/ml, an effective antibiofilm concentration[18]) in the presence of 1% of $H_2O_2$ (v/v). Remarkably, the combination of Fer with $SnF_2$ has substantially greater antibiofilm activity than the combination of Fer and NaF (Supplementary Fig. 1c, d), resulting in no detectable viable bacteria and near complete biomass abrogation. In view of this result, we hypothesized that $SnF_2$ might be interacting with Fer for enhanced bioactivity.

We then investigated antibiofilm activity using two dose-response studies with varying concentrations of Fer and $SnF_2$. Given the potency of this combination and of high fluoride (1000 ppm of F) concentration, we used the lowest dosage of $SnF_2$ (250 ppm of F) known to provide therapeutic effect as the upper fluoride dose limit. First, a fixed dose (1 mg of Fe/ml) of Fer was mixed with various concentrations of $SnF_2$ (0–250 ppm of F), and the number of viable cells and biomass were determined. As expected, Fer displayed a strong antibacterial effect against *S. mutans* biofilm (>3-log reduction of viable cells; Fig. 2a), while also reducing biomass (Fig. 2b). When Fer was mixed with increasing concentrations of $SnF_2$, both the antibacterial activity and the inhibitory effect on the biomass enhanced in a dose-dependent manner, indicating that $SnF_2$ can help improve the antibiofilm efficacy of Fer. Next, the Fer concentration was varied (0–1 mg of Fe/ml) while using a constant $SnF_2$ dose (250 ppm of F). When

combined with Fer, the antibacterial effect of $SnF_2$ increased in a dose-dependent manner, resulting in >5-log reduction of viable cells compared to control when the concentration of Fer reached 1 mg of Fe/ml. Notably, the combination of Fer and $SnF_2$ was at least 2500-fold more effective in killing *S. mutans* cells than $SnF_2$ alone (Fig. 2c), suggesting a synergistic effect. We also found that $SnF_2$ (250 ppm of F) substantially reduces biomass (Fig. 2d), although inclusion of increasing amounts of Fer did not enhance the bioactivity. The reduction of dry biofilm mass in response to $SnF_2$ treatment is likely due to the inhibition of secreted glucosyltransferases that are integral to the production of exopolysaccharides (EPS) by *S. mutans*, as reported by others[26].

To further assess the antibiofilm activity of the combination of Fer (1 mg of Fe/ml) and $SnF_2$ (250 ppm of F), confocal imaging was performed using fluorescent labeling of the bacterial cells and α-glucan EPS. As depicted in Fig. 2e, the control biofilm contained bacterial clusters (in green) spatially arranged with abundant EPS (in red) matrix forming a densely packed structure. In a sharp contrast, the combination of Fer and $SnF_2$ impaired the accumulation of biofilm where only small cell clusters with sparsely distributed EPS were detected. The orthogonal view images revealed that the spatial distribution of bacteria and EPS across the biofilm thickness was substantially compromised in the combination treated biofilm. These findings were further confirmed by quantitative computational analyses, which showed that the combination of Fer and $SnF_2$ markedly reduced the biovolume of bacterial cells (Fig. 2f) and EPS (Fig. 2g).

### $SnF_2$ stability in combination with Fer

Given the enhanced efficacy of the combination of Fer and $SnF_2$, we investigated the physicochemical properties of this combination. The hydrodynamic diameter of Fer did not change significantly after adding $SnF_2$ (Supplementary Table 1), indicating that $SnF_2$ is stable in solutions of Fer. Additionally, we noticed that the zeta potential of Fer (Supplementary Table 2) became less negative, serving as an initial evidence that Fer interacts with $SnF_2$, since coordination of $Sn^{2+}$ by the carboxylate groups is expected to lower the charge density of the carboxymethyl-dextran (CMD) corona. Representative transmission electron microscopy (TEM) of Fer and Fer + $SnF_2$ after 1 h incubation in 0.1 M sodium acetate buffer (pH 4.5) are presented in Supplementary Fig. 2. Consistent with dynamic light scattering (DLS) data (Supplementary Table 1), mixing Fer with $SnF_2$ did not affect the size of Fer.

It is noteworthy that $SnF_2$ has limited stability in aqueous solutions owing to its high susceptibility to hydrolysis and oxidation[27,28] requiring chemical additives (e.g., chelating agents) or anhydrous formulation[20], which can reduce fluoride bioavailability. We unexpectedly found that $SnF_2$ was stable in aqueous solutions containing Fer without additives. To further investigate the stability of $SnF_2$ in the presence of Fer, $SnF_2$ (250 ppm of F) was mixed with increasing amounts of Fer (0.25–1 mg of Fe/ml) in 0.1 M sodium acetate buffer at pH 4.5. We observed that the solution containing $SnF_2$ mixed with Fer was limpid after 24 h in sodium acetate buffer at pH 4.5 (Fig. 3a), demonstrating that Fer can enhance the stability of $SnF_2$.

The enhanced stability of $SnF_2$ in the presence of Fer motivated us to investigate their chemical interactions. The core of Fer is coated with carboxymethyl-dextran (CMD)[29]. Thus, we explored whether $SnF_2$ can interact with CMD. $SnF_2$ alone or mixed with CMD was incubated in 0.1 M sodium acetate buffer (pH 5.5) for 24 h. We observed immediate formation of a precipitate when $SnF_2$ was dissolved in sodium acetate buffer, whereas the solution was limpid when mixed with CMD even after 24 h incubation (Fig. 3b). In addition, the mixture of $SnF_2$ and CMD was characterized by $^1H$ nuclear magnetic resonance (NMR) spectroscopy (Fig. 3c). In the CMD spectrum, the anomeric proton (H1) in the C1 position was identified at 4.9 ppm, and protons (H2–H6) at the C2–C6 positions were detected at 3.2–4.0 ppm. The peak at 4.0–4.2 ppm (denoted as "a") is attributed to the protons of the carboxymethyl moieties as determined previously[30]. After CMD was

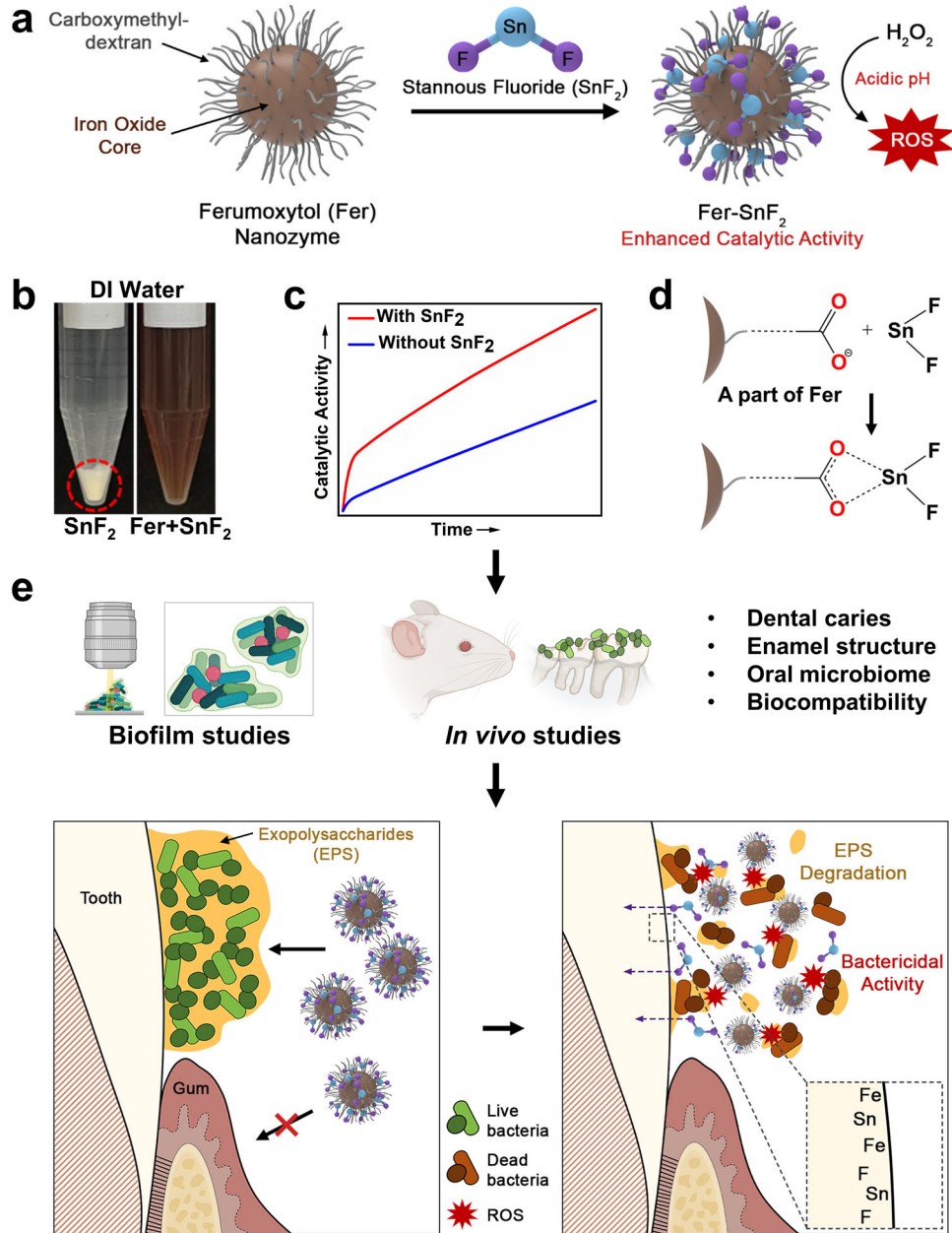

**Fig. 1 | Chemical interactions and therapeutic activity of the combined treatment of Fer and SnF₂.** Fer chemically interacts with SnF₂ (**a**) to enhance stability in aqueous solution without any additives (the red circle highlights precipitate) (**b**) while boosting catalytic activity (**c**). **d** Further biochemical and spectrometry analyses reveal that Sn²⁺ is bound by carboxylate groups in the carboxymethyl-dextran coating of Fer. **e** Using laboratory and in vivo models, we find synergistic activities to enhance bioactivity against biofilms and caries-protective effects (without increasing fluoride exposure), while co-delivering fluoride, iron, and tin on the outer enamel surface (forming a polyion film) without deleterious effects on oral tissues and the microbiota.

mixed with SnF₂, a clear shift in peak "a" was observed when compared to that of CMD alone. This suggests that Sn²⁺ binds to the carboxymethyl moieties of CMD, which may account for the enhanced stability of SnF₂ with Fer. Note that similar ¹H NMR studies of SnF₂ and Fer are not possible due to the superparamagnetism of Fer interfering with ¹H NMR measurements.

In order to further investigate the effects of CMD on SnF₂ stability, we compared it with several control materials, i.e. dextran (Dex) (a similar polymer to CMD, but without carboxylic acid groups), as well as citric acid (CA), L-ascorbic acid (AA), and poly(acrylic acid) (PAA), which are all entities that all contain carboxylic acid groups. We found that Dex did not enhance the stability of SnF₂, whereas each material that contains carboxylic acid groups did enhance stability (Fig. 3d–f, Supplementary Fig. 3). The Fer

formulation also contains mannitol (Man)[29], which is an antioxidant[31]. Since antioxidants can prevent the oxidation of SnF₂[32], we added SnF₂ to various amounts of Man (1–10 mg/ml). Surprisingly, we did not observe any noticeable change in the stability of SnF₂ even with excess amounts of Man (10 mg/ml) (Supplementary Fig. 4), implying that Man does not have any noticeable effect on enhancing the stability of SnF₂. All of these findings suggest that Sn²⁺ is bound to carboxylate groups, thereby enhancing the stability of SnF₂ when mixed with Fer.

### Catalytic activity of Fer in combination with SnF₂

To explore whether SnF₂ could influence the catalytic activity of Fer, we used the 3,3′,5,5′-tetramethylbenzidine (TMB) colorimetric assay for peroxidase-like activity following a previously published

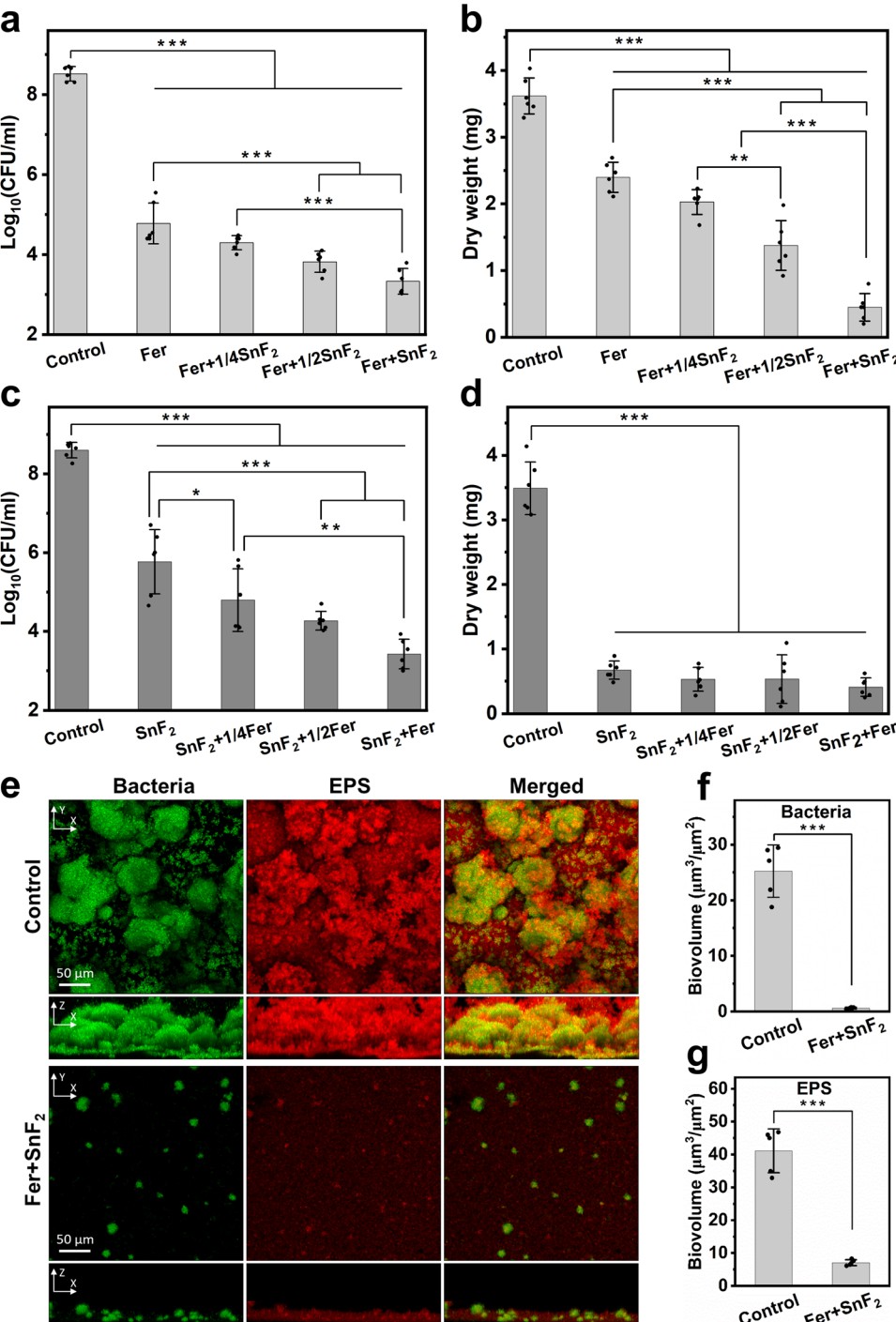

**Fig. 2 | Antibiofilm studies of the combinations of Fer and SnF₂. a, b** The effect of different concentrations of Fer (1 mg of Fe/ml) and SnF₂ (0–250 ppm of F) on the bacterial viability (**a**) and the mass of biofilm (**b**) after 43 h. 1/4SnF₂, 1/2SnF₂, and SnF₂ indicate 62.5 ppm of F, 125 ppm of F, and 250 ppm of F, respectively, (*n* = 6, 3 independent experiments with two replicates). The effect of different concentrations of Fer (0-1 mg of Fe/ml) and SnF₂ (250 ppm of F) on the bacterial viability (**c**) and the mass of biofilm (**d**) after 43 h. 1/4Fer, 1/2Fer, and Fer indicate 0.25 mg of Fe/ml, 0.5 mg of Fe/ml, and 1 mg of Fe/ml, respectively, (*n* = 6, 3 independent experiments with two replicates). **e** Confocal microscopy images of biofilms with or without treatment with Fer (1 mg of Fe/ml) and SnF₂ (250 ppm of F) after 43 h. Bacterial cells were stained with SYTO 9 (in green), and EPS was labeled with Alexa Fluor 647-dextran conjugate (in red). Quantitative analysis of biovolume of bacterial cells (**f**) and EPS (**g**) in the biofilm with or without Fer + SnF₂ (analyzed using COMSTAT2; *n* = 5, taken 5 random images from three independent experiments). All biofilms except the control group were treated with H₂O₂ (1%, v/v) for 5 min at the end of the experimental period (43 h) before the analysis. The data are presented as mean ± standard deviation. *$p < 0.05$, **$p < 0.01$, ***$p < 0.001$; one-way ANOVA followed by Tukey test. Source data are provided as a Source Data file.

protocol[33], with some modifications. TMB is a chromogenic compound that yields a blue color upon oxidation with an absorption peak at 652 nm in the presence of reactive oxygen species (ROS), such as hydroxyl radical (•OH)[34]. As shown in Fig. 4a, b, SnF₂ alone did not produce a noticeable amount of ROS. In contrast, the catalytic activity of Fer increased significantly after combining with SnF₂ as demonstrated by increased colorimetric reaction (Fig. 4a, b), suggesting that SnF₂ enhanced the catalytic activity of Fer. Photographs in the inset of Fig. 4a exhibit the color change in each condition (SnF₂, Fer, and Fer + SnF₂ from left to right, respectively).

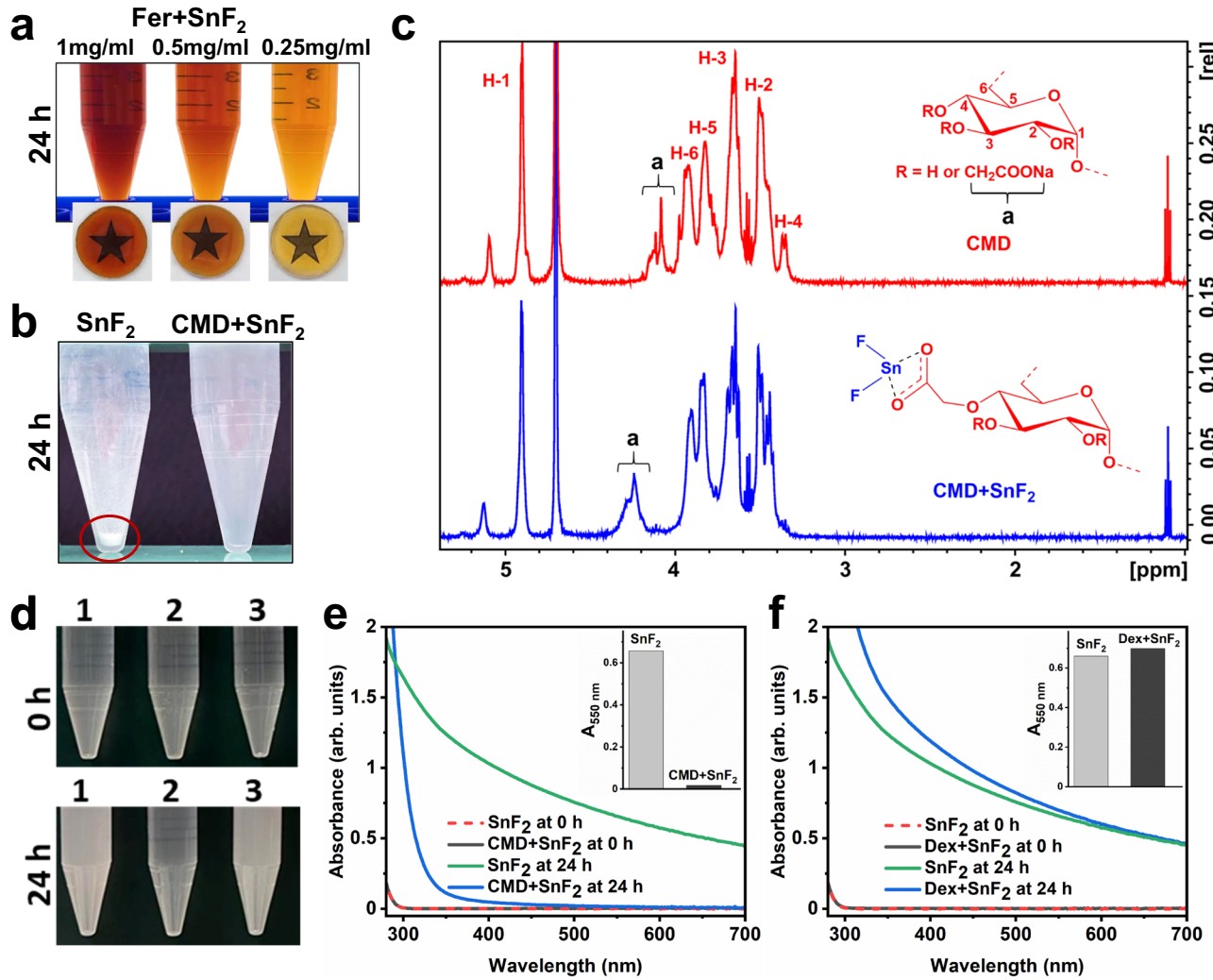

**Fig. 3 | Enhanced stability of SnF$_2$ in the presence of Fer. a** Photographs of combinations of SnF$_2$ and differing concentrations of Fer (0.25–1 mg of Fe/ml) at pH 4.5 (0.1 M sodium acetate buffer) after 24 h incubation. **b** Photographs of SnF$_2$ and the combination of CMD and SnF$_2$ at pH 5.5 (0.1 M sodium acetate) after 24 h incubation. The red circle highlights precipitate. **c** $^1$H NMR spectra of CMD and CMD + SnF$_2$, 'a' peaks represent the protons of carboxymethyl groups. **d** Photographs of SnF$_2$ in different conditions at pH 4.5 (0.1 M sodium acetate buffer). The samples are: 1. SnF$_2$ alone, 2. CMD + SnF$_2$, and 3. dextran (Dex) + SnF$_2$. Top: 0 h; bottom: after 24 h. **e** UV–visible absorption spectra of SnF$_2$ (250 ppm of F) with or without CMD (1 mg/ml) at pH 4.5 (0.1 M sodium acetate buffer) after 0 or 24 h incubation. The inset of (**e**) shows the absorbance of SnF$_2$ and CMD + SnF$_2$ at 550 nm after 24 h incubation as a measure of turbidity. **f** UV–visible absorption spectra of SnF$_2$ (250 ppm of F) with or without Dex (1 mg/ml) at pH 4.5 (0.1 M sodium acetate buffer) after 0 or 24 h incubation. The Inset of (**f**) shows the absorbance of SnF$_2$ and Dex + SnF$_2$ at 550 nm after 24 h incubation as a measure of turbidity. Source data are provided as a Source Data file.

Notably, we found that the enhancement of ROS production in the presence of SnF$_2$ is dependent on pH, concentration, and incubation time. The highest catalytic activity was observed at pH 4.5 (Fig. 4c). The greater ROS production at acidic pH conditions (characteristic of pathological conditions associated with dental caries) and the minimal ROS generation close to neutral (physiological) pH suggests a selectivity toward pathogenic bacteria. Given the pH-dependent activity, we also tested binding interactions between SnF$_2$ and Fer at different pH values and found that the highest co-binding occurred at pH 4.5 (Supplementary Fig. 5). Surprisingly, a very small amount of SnF$_2$ (0.156 μg/ml) is sufficient for enhancing the catalytic activity of Fer (Supplementary Fig. 6), and the ratio of the mass of SnF$_2$ to the mass of Fe is determined to be 0.0078:1 (under the present experimental conditions). We observed that more ROS can be detected in the presence of SnF$_2$ within 10 min incubation (Supplementary Fig. 7a) gradually increasing to reach the highest level at 6 h (Supplementary Fig. 7b), which was maintained for prolonged period.

To further confirm the enhancement of the peroxidase-like activity of Fer in the presence of SnF$_2$, we employed a multi-pronged approach. First, we used o-phenylenediamine (OPD), a colorless substrate, which yields an oxidized product with a characteristic yellow color when reacting with ROS with an absorption peak at 450 nm[35]. As expected, the catalytic activity of Fer increased markedly after adding SnF$_2$ as compared to Fer alone and SnF$_2$ alone (Fig. 4d). We also measured ROS production via a photoluminescence (PL) method using 2′,7′-dichlorofluorescin diacetate (DCFH-DA) as a ROS tracking indicator. DCFH-DA (a nonfluorescent molecule) yields a fluorescent molecule DCF in the presence of ROS[36]. As depicted in Fig. 4e, the PL intensity increased to a greater extent after combining Fer with SnF$_2$. Then, we measured the amount of hydroxyl radical (•OH) using coumarin as a photoluminescent probe molecule[37,38]. As seen in Fig. 4f, Fer and SnF$_2$ in combination generated significantly more •OH than Fer alone, further demonstrating that SnF$_2$ enhanced the catalytic activity of Fer. In contrast, SnF$_2$ alone did not produce a noticeable amount of •OH (Supplementary Fig. 8), consistent with its lack of catalytic activity.

Next, we investigated whether the augmented catalytic activity arises from different fluoride or stannous salts. We replaced SnF$_2$ with NaF, a commonly used fluoride salt in oral care formulations, or barium

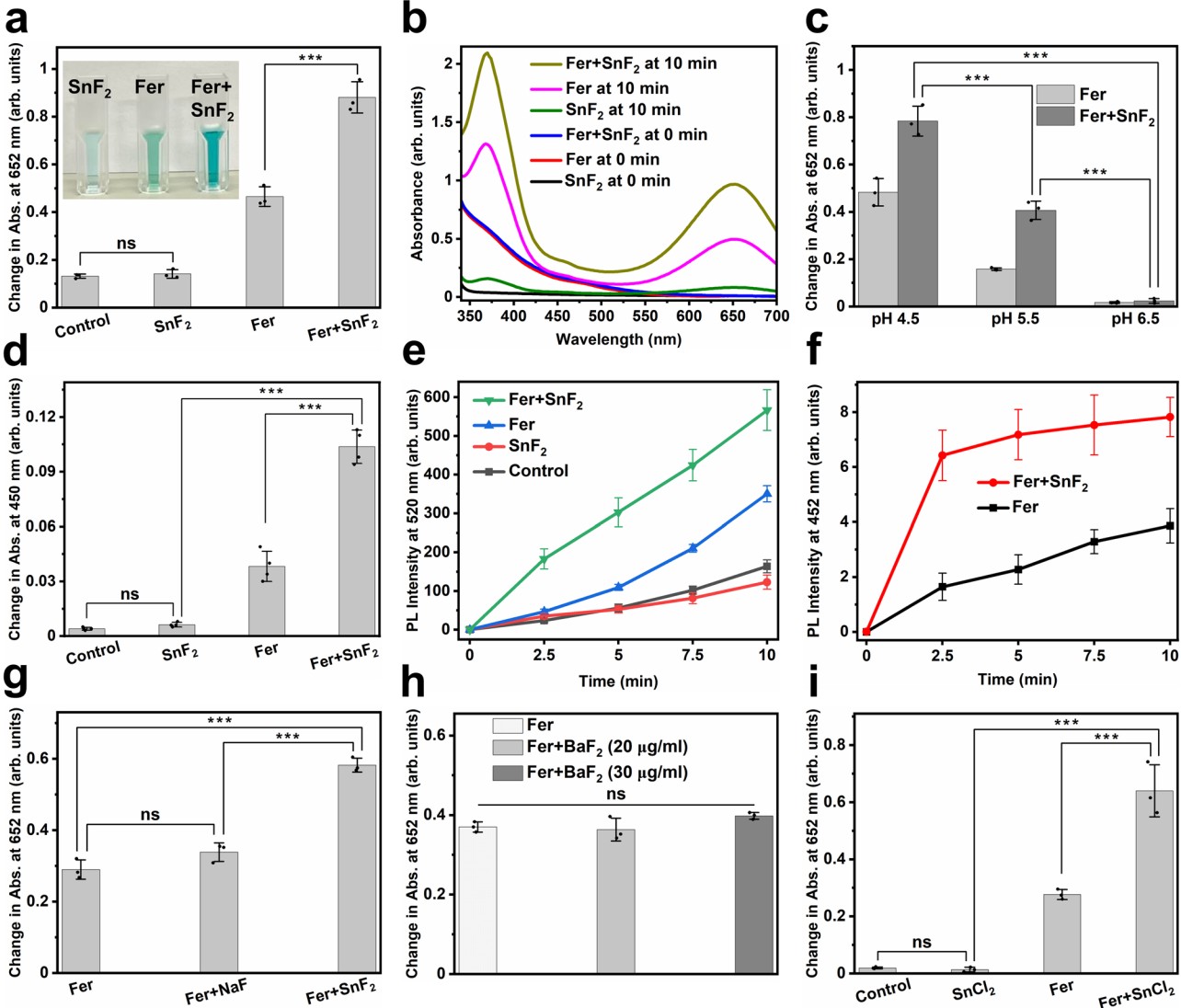

**Fig. 4 | Enhanced catalytic activity of Fer in the presence of SnF₂. a** Change in the absorption of TMB (chromogenic substrate) at 652 nm in different conditions. Inset: Photographs of TMB incubated with various reagents (left: SnF₂ alone, middle: Fer alone, and right: Fer + SnF₂) 10 min after H₂O₂ addition ($n = 3$ independent experiments). **b** UV−visible absorption spectra of TMB in the presence of SnF₂, Fer, or Fer + SnF₂ at the times indicated. **c** Peroxidase-like activity of Fer and Fer + SnF₂ at three pH values (4.5, 5.5, and 6.5) as determined by the colorimetric TMB assay ($n = 3$ independent experiments). **d** Change in the absorption of OPD at 450 nm in different conditions ($n = 4$ independent experiments). The increase in absorption at 450 nm shows ROS production. **e** Comparison of change in PL intensities of DCF at 520 nm at various conditions ($n = 5$ independent experiments). The increase in PL intensity at 520 nm depicts ROS production. **f** The change in PL intensity of 7-hydroxycoumarin at 452 nm as a function of time in the presence of Fer with or without SnF₂ ($n = 5$ independent experiments). The increase in the PL intensity at 452 shows the generation of •OH. Effect of NaF (**g**), BaF₂ (**h**), and SnCl₂ (**i**) on the catalytic activity of Fer in 0.1 M sodium acetate buffer (pH 4.5) ($n = 3$ independent experiments). The data are presented as mean ± standard deviation. ***$p < 0.001$; ns nonsignificant; one-way ANOVA followed by Tukey test. Source data are provided as a Source Data file.

fluoride (BaF₂), another fluoride salt with a divalent cation of comparable size to Sn²⁺. We found that neither NaF nor BaF₂ increased the catalytic activity of Fer noticeably (Fig. 4g, h), suggesting that F ions may not play a crucial role in enhancing the ROS production performance of Fer. Conversely, we used SnCl₂ to evaluate whether Sn ions play a role in strengthening the ROS generation capability of Fer. We found that SnCl₂ enhanced the catalytic activity of Fer (Fig. 4i), indicating that Sn ions may be playing a dominant role in increasing the catalytic performance of Fer. Taken together, these findings support that SnF₂ can boost the catalytic ability of Fer, indicating that Fer and SnF₂ combination is an effective ROS-generating therapy that can target biofilms under pathological (acidic) conditions.

We examined whether Fer released iron ions when combined with SnF₂ using inductively coupled plasma optical emission spectroscopy (ICP-OES). As depicted in Supplementary Fig. 9a, the presence of SnF₂ slightly increased iron ions release from Fer at acidic pH (4.5). It is noteworthy that the amount of leached irons from Fer + SnF₂ formulation at circumneutral pH is negligible (Supplementary Fig. 9b). Conversely, the iron leached from Fer + SnF₂ at acidic pH values could provide an added benefit. Iron ions have shown cariostatic effects as they can precipitate on the surface of enamel and promote the adsorption of phosphate and calcium ions, thereby reducing enamel demineralization[39,40].

It has been reported that H₂O₂ can induce the oxidation of SnF₂ as H₂O₂ is a well-known oxidizing agent[41–43]. To assess the stability of SnF₂ when mixed with Fer after the catalytic reaction, we conducted a series of experiments. Initially, we recorded UV−visible absorption spectra of SnF₂ in the presence and absence of H₂O₂. As expected, the absorption

curve of $SnF_2$ increased in the presence of $H_2O_2$ when compared to $SnF_2$ alone (Supplementary Fig. 10a), indicating that $H_2O_2$ caused the oxidation of $SnF_2$. However, the absorption spectra of Fer + $SnF_2$ remained almost unchanged after the catalytic reaction (in the presence of $H_2O_2$), even after 60 min of catalysis (Supplementary Fig. 10b) as compared to Fer alone, suggesting that Fer may prevent the oxidation of $Sn^{2+}$. We also compared the absorbance of $SnF_2$ with or without the presence of Fer at 550 nm as a quantitative measurement of turbidity after 1 h incubation with $H_2O_2$. As shown in Supplementary Fig. 11, the presence of Fer significantly reduced the oxidation of $SnF_2$.

To further evaluate the stability of $SnF_2$ when mixed with Fer after $H_2O_2$ treatment, we measured the concentration of free tin ions with or without $H_2O_2$ exposure by ICP-OES (Supplementary Fig. 12). We found that the amount of free tin ions was slightly less after the catalytic reaction (i.e., $H_2O_2$ treatment), thereby indicating minimal loss of $SnF_2$ stabilization. Based on these experiments, it is apparent that the majority of $SnF_2$ remains bound on Fer and stable even after the catalytic reaction. These findings provide further evidence that Fer acts as a stabilizing agent for $SnF_2$, effectively reducing its oxidation in the presence of $H_2O_2$.

Altogether, the increased stability of $SnF_2$ in aqueous solutions is mediated at least in part via interactions with CMD, which may be important for both fluoride bioavailability and fluoride delivery. Unexpectedly, the presence of $SnF_2$ boosts the ROS generation capability of Fer at acidic pH, thus enhancing antibiofilm efficacy under pathological condition. This synergistic Fer and $SnF_2$ combination provide a potent yet pH-dependent ROS-based therapy with enhanced antimicrobial fluoride stability that could prevent the onset of dental caries in vivo.

### Biocompatibility of Fer + $SnF_2$ in vitro
To examine whether this combined treatment is viable for use in vivo, the cytotoxicity of the combination of Fer and $SnF_2$ was assessed in immortalized human gingival keratinocytes (HGKs) using MTS assay. The cells were incubated with the combination of Fer (1 mg of Fe/ml) and $SnF_2$ (250 ppm of F) for 10 min, followed by 24 h incubation with fresh cell culture media. We found that the combined treatment of Fer and $SnF_2$ had no adverse effect on cell viability (Supplementary Fig. 13).

### Impact of Fer/$SnF_2$ on caries development and on enamel surface in vivo
Topical applications of Fer and $SnF_2$ in vivo were assessed using a rodent model that mimics the characteristics of severe human caries[44], including a sugar-rich diet and the development of carious surface zones[45]. In this model, rat pups were infected with *S. mutans* UA159 (a cariogenic oral bacterium) and provided sucrose-containing food and water (Fig. 5a). By feeding sugar-rich diet, tooth enamel progressively develops caries lesions (analogous to those observed in humans), proceeding from initial areas of demineralization to severe lesions characterized by enamel structure damage and cavitation (Fig. 5b). The test agents were topically applied twice daily with 1 min exposure time (Fig. 5a) to mimic the clinical use of a mouthwash. After the experimental period, the incidence and severity of caries lesions were evaluated. We also included a reduced concentration of the combination of Fer (0.25 mg of Fe/ml) and $SnF_2$ (62.5 ppm of F), since these lower amounts were capable of significantly killing the bacteria ($p < 0.001$) and reducing biomass ($p < 0.001$) compared to control in vitro (Supplementary Fig. 14).

Quantitative caries scoring analyses revealed that the treatment of Fer in combination with $SnF_2$ was exceptionally effective in preventing caries development with higher efficacy than either alone ($p < 0.001$) (Fig. 5c). It nearly abrogated caries initiation and completely blocked further caries lesions development, thus preventing the onset of cavitation altogether (Fig. 5d, e). The efficacy of the lower dosage of Fer and $SnF_2$ treatment was significantly greater than the control group

($p < 0.001$), and as effective as Fer (1 mg of Fe/ml) or $SnF_2$ (250 ppm of F) treatment alone. This demonstrates that the combination of Fer and $SnF_2$ has a synergistic effect for efficient biofilm treatment in vivo. Herein, we used 1% of $H_2O_2$ based on our previous dose-response studies[18,33,46]. As $SnF_2$ enhances the catalytic activity of Fer, lower concentrations of $H_2O_2$ may also be effective in combination with Fer and $SnF_2$ for biofilm disruption and caries prevention.

To determine the impact of treatment on the elemental composition of the enamel surface, lamellae oriented normal to the external enamel surface (EES) of rat mandibular first molars (M1) were lifted out using a conventional focused ion beam (FIB) technique (Supplementary Fig. 15). Line profiles normal to the EES were determined by scanning transmission electron microscopy (STEM) with energy dispersive spectroscopy (STEM-EDS) (Fig. 5f) and aligned to the outer surface (see Methods). Comparing M1 from Fer + $SnF_2$ and control groups (Fig. 5f(i)), we find that the combined mole fractions of Fe, Sn, and F are substantially elevated, and the sum of the mole fractions Ca and P correspondingly reduced, in a thin film at the surface. Preliminary analyses revealed that the thickness of this film varies from ~50 to greater than 300 nm. Inspection of single element profiles from a 50 nm-thick film (Fig. 5f(ii–vi)) reveals that while the calcium mole fraction is reduced from ~25 at% to less than 5 at% in this layer, the mole fraction of oxygen remains at the same level as in the underlying enamel (Fig. 5f(iii)). Sn reaches slightly more than 10 at% (Fig. 5f(vi)), while Fe is closer to 7 at% (Fig. 5f(v)). While the profiles of the latter show broad maxima in the center of the layer, F levels appear highest at the outer surface at ~5 at% and decline to ~1 at% at the interface (Fig. 5f(iv)). The presence of a Fe/Sn/F-rich layer was confirmed on a separately prepared sample from the same, treated rat molar using STEM with electron energy loss spectroscopy (STEM-EELS) (Supplementary Fig. 16). Furthermore, the presence of Fe, Sn, and F at the surface of treated teeth, and the absence of these ions on vehicle-treated controls was confirmed using X-ray photoelectron spectroscopy (XPS, Supplementary Fig. 17).

In high-resolution TEM (HRTEM) images, the Fe/Sn-rich layer was apparent as a slightly darker band between the enamel and the protective layers of FIB-deposited carbon and FIB-deposited Pt/C (Supplementary Fig. 18). Lattice fringes were readily apparent in enamel (Supplementary Fig. 18a), and Fast Fourier transform (FFT) images and radial integrals (Supplementary Fig. 18b, e) revealed sharp features consistent with the {002}, {3$\bar{2}$1}, and {3$\bar{3}$0} sets of planes of crystalline hydroxylapatite. The Fe/Sn-rich layer did not display lattice fringes, and FFT images only showed diffuse scattering with a broad maximum at ~0.33 $nm^{-1}$ (Supplementary Fig. 18c, f), consistent with an amorphous oxide layer.

Taken together, there is strong evidence that there is a layer comprised of Fe and Sn with varying amounts of fluoride present at the surface of the teeth of animals treated with Fer + $SnF_2$. The presence of Ca and P in this layer may indicate co-precipitation during its formation. Gradients of Fe, Sn, and F at the interface between the film and the underlying enamel suggest that these ions diffuse into enamel, but this remains to be confirmed.

### Effect of Fer/$SnF_2$ on host microbiota and oral tissues in vivo
The effects of Fer and $SnF_2$ on oral microbiota and surrounding soft tissues were also evaluated to assess the impact on oral microbiome diversity and oral tissue toxicity. All treatment groups showed no significant differences in alpha diversity between each group (Fig. 6a, b; $p > 0.05$, Willcox test). Furthermore, weighted UniFrac distances analyzed from principal coordinate analysis (PCoA) by treatment groups revealed that Fer and $SnF_2$ treatment group has a similar composition with the lowest dispersion (Fig. 6c, green dots), indicating no deleterious effects on the oral microbiota diversity ($p > 0.05$, PERMANOVA). Notably, microbiome data revealed a higher abundance of acidogenic bacterial genera such as *Streptococcus* and *Lactobacillus* in the control

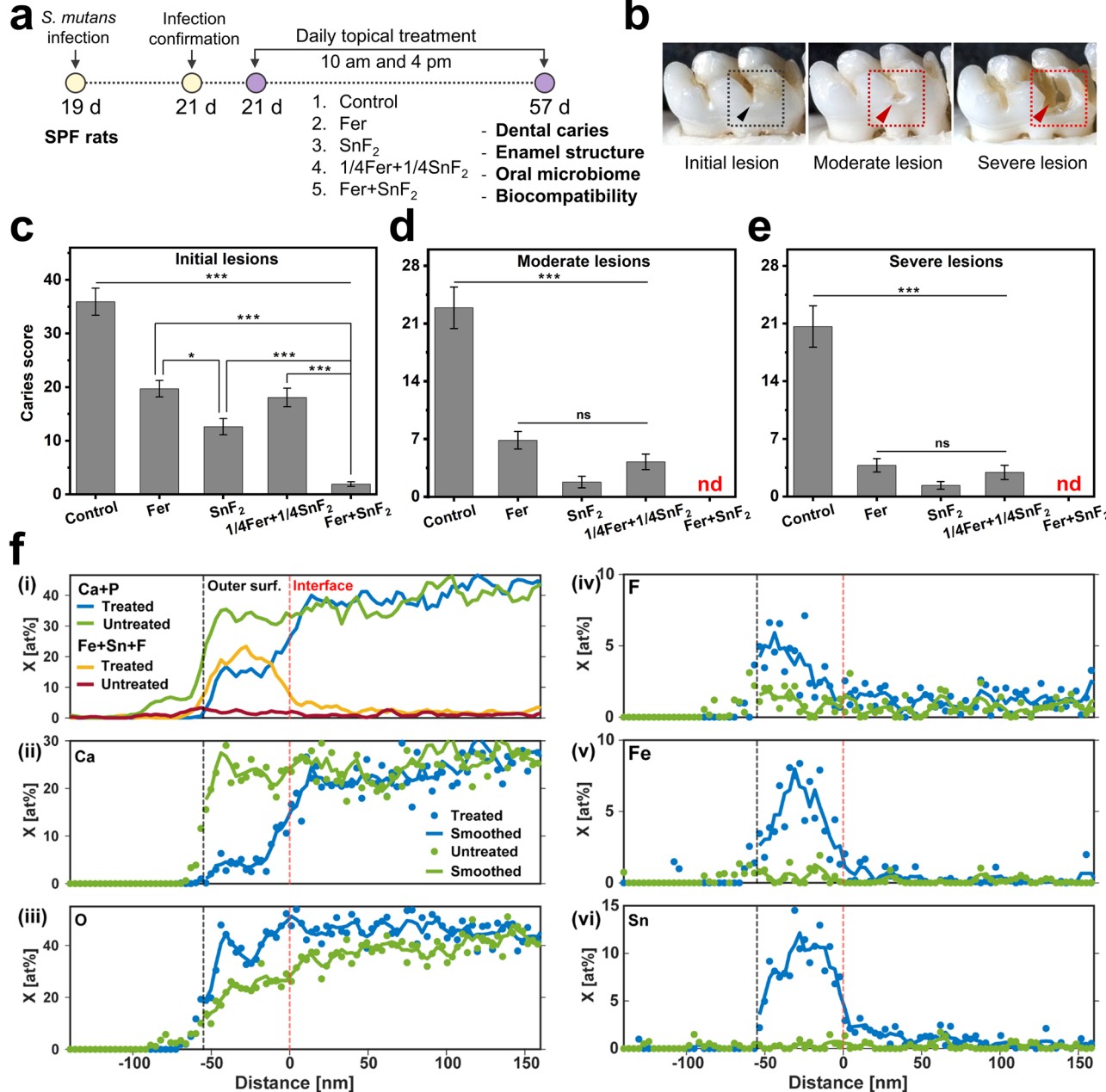

**Fig. 5 | Therapeutic efficacy of the combination of Fer and SnF$_2$ against dental caries in vivo.** Caries development was evaluated on all tooth surfaces, while elemental composition of the surface of M1 molars of rats treated with Fer + SnF$_2$, and from untreated controls, was assessed by STEM-EDS. **a** Experimental design of the in vivo study ($n = 16$, for each treatment group). **b** An illustration showing an initial lesion (enamel affected), a moderate lesion (dentin affected), and a severe lesion (full cavitation). **c–e** Caries scores recorded from smooth surface. The caries scores were recorded according to Larson's modification of Keyes' scoring system for stages and extent of carious lesion severity. Fer, 1/4Fer, SnF$_2$, and 1/4SnF$_2$ indicate 1 mg of Fe/ml, 0.25 mg of Fe/ml, 250 ppm of F, and 62.5 ppm of F, respectively. The

data are presented as mean ± standard error of the mean. *$p < 0.05$, ***$p < 0.001$; ns nonsignificant, nd nondetectable; one-way ANOVA followed by Tukey test. **f** Plot of the sum of mean mole fractions for Ca and P, and for Fe, Sn, and F (**i**), and plots of mean mole fraction for Ca (**ii**), O (**iii**), F (**iv**), Fe (**v**), and Sn (**vi**) vs. distance in the direction normal to the EES. Profiles were manually aligned on the outer surface, and the distance axis is referenced to the approximate position of the interface between the Fe/Sn/F-rich layer and the underlying enamel of the treated sample. For (**ii**)–(**vi**), solid circles indicate the mean mole fraction at a given distance, and lines indicate the moving average of the mole fraction with span 3 (denoted as "smoothed" in legend). Source data are provided as a Source Data file.

group, whereas they decreased in all the treatments (Fig. 6c, d). *Veillonella* is known to consume acids produced by other acidogenic oral bacteria to grow and survive. *Veillonella* is especially reduced in the combined treatment groups, i.e., 1/4Fer + 1/4SnF$_2$ and Fer + SnF$_2$ (Fig. 6c, d), indicating reduced acidogenic environment. In contrast, commensal genera related to oral health such as *Haemophilus* and *Rothia* were increased in treatment groups. The treatment can affect the localized acidic microenvironment of plaque biofilm by modulating

the growth of oral health-associated bacteria. *Rothia* is a nitrate-reducing oral bacteria that can generate nitrite in proximity to raise the local pH[47]. In Fer + SnF$_2$ treatment, Sn-bound in the vicinity could serve as electron donors, facilitating nitrite and ammonia production by *Rothia*[48]. Furthermore, this localized pH change may act as a triggering factor shifting relative abundance between *Streptococcus/Lactobacillus* (as acidogenic bacteria) and *Haemophilus*[49]. Bacterial shifts of lactate-producing *Streptococcus/Lactobacillus* may also affect the abundance

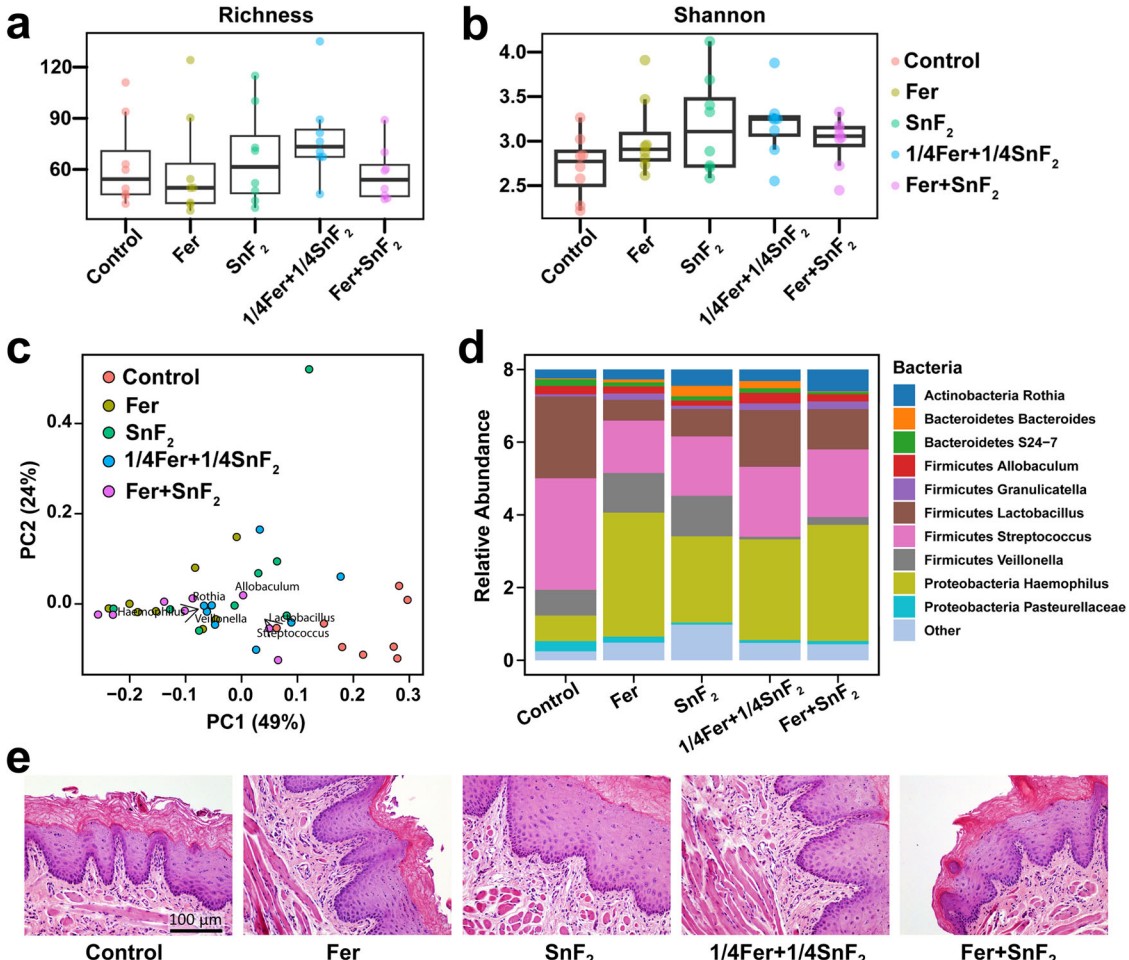

**Fig. 6 | Effects of Fer and SnF₂ on oral microbiome and gingival tissue in vivo post-treatment. a**, **b** Alpha diversity measured by Richness and Shannon indexes shows no significant differences between groups. The lines on the boxplots represent the lower and upper quartiles, while the center line indicates the median ($n = 8$, for each treatment group). **c** Principal coordinate analysis (PCoA) using weighted UniFrac distances reveal that the Fer + SnF₂ group has a similar composition and the lowest distances between samples. **d** The bar plot shows the main bacterial genera found across all samples, distributed by treatment groups ($n = 8$, for each treatment group). **e** Histology of the gingival tissue with treatments noted. Fer, 1/4Fer, SnF₂, and 1/4SnF₂ indicate 1 mg of Fe/ml, 0.25 mg of Fe/ml, 250 ppm of F, and 62.5 ppm of F, respectively.

of lactate-utilizing bacteria, such as *Veillonella*, as observed in the treatment group. Taken together, the microbiome data indicates that bacterial diversity is not affected as a community (Fig. 6d), but specific bacteria associated with pathogenic environments are reduced by the combination treatment. In the current study, we could not monitor *S. mutans* using a partial region of the 16S rRNA gene. Moreover, *Streptococcus* species have poor taxonomic resolution below the genus level[50]. Future studies using shotgun whole genome sequencing are warranted for monitoring functional microbiome changes at species-level with higher resolution and accuracy as well as microbial association network analyses to identify interspecies interactions.

Histopathological analysis of gingival tissues revealed no indication of an acute inflammatory response, cytotoxicity, necrosis, or any changes in vascularization or proliferation, suggesting that the Fer and SnF₂ treatment is biocompatible (Fig. 6e), consistent with in vitro data. Collectively, the data show that the combination was substantially more potent than either alone, whereas a lower concentration of agents in combination was as effective as each alone at full strength, indicating a synergistic effect between Fer and SnF₂. In addition, the treatments did not disrupt the ecological balance of the oral microbiota or cause deleterious effects on the surrounding host tissues, indicating high precision for targeting cariogenic plaque-biofilms and preventing disease progression in vivo.

## Discussion

In summary, we unexpectedly find a remarkable synergy between ferumoxytol (Fer) nanozymes and stannous fluoride (SnF₂) in potentiating antibiofilm and anticaries efficacy, which is particularly relevant given that current treatments are insufficient for controlling biofilm and preventing demineralization simultaneously in high-risk populations prone to disease. The combination treatment is far more effective than either alone and completely halts the progression of caries lesions and cavitation in a rodent model, without adverse effects on the surrounding host tissues or on the oral microbiota diversity in vivo. Notably, we observed initial enamel lesions and eventually cavity formation when treated with SnF₂ or Fer alone, indicating that sufficient acid is still generated to attack enamel. In sharp contrast, early lesions were seldom seen when treated with SnF₂ in combination with Fer. Furthermore, comparable therapeutic effects were achieved even at 4 times lower fluoride concentration (62.5 ppm of F) when mixed with Fer, demonstrating the possibility of a therapy that uses very low doses (typical amounts in oral formulations ranges from 1000 to 1500 ppm of F). Such therapeutic synergy has not been observed previously in this animal model that mimics severe disease. Further analyses revealed that the improved effects achieved with the combination system can be attributed to three factors: (i) stabilization of SnF₂ through tin-carboxylate interaction, (ii) significant enhancement of the

catalytic activity of Fer, and (iii) formation of a Fe/Sn/F-rich film at the outer surface of the tooth enamel. These properties acting in concert can potentiate antibiofilm activity and enhance enamel resistance against demineralization, while also displaying therapeutic effect at lower dosages.

It is well known that $SnF_2$ undergoes oxidation and hydrolysis in aqueous solutions. Therefore, maintaining the stability and efficacy of $SnF_2$ has been a ubiquitous challenge since the inception of the use of $SnF_2$ in the 1950s. Several formulations have been developed to enhance the stability of $SnF_2$ in oral care products, including (i) removal of water from the formula or use of low water content, (ii) addition of extra stannous salts, such as $SnCl_2$, as a sacrificial source of stannous ions, and (iii) addition of complexing agents, such as zinc phosphate[20]. However, the currently available methodologies have some limitations, including interactions of stannous complexes with other ligands present in toothpaste that affect the therapeutic efficacy of $SnF_2$[42]. Herein, we offer an alternate and facile approach of stabilizing $SnF_2$ in aqueous solutions with additional therapeutic benefits. Specifically, we show (i) Fer can stabilize $SnF_2$ via simple mixing without using any excipient ingredients (and effectively reducing its oxidation in the presence of $H_2O_2$) while (ii) enhancing anticaries effect of $SnF_2$ when mixed with Fer. We find that Sn binding through the carboxylate groups of the Fer formulation contributes to the stabilization of $SnF_2$ and also plays a role in enhancing the catalytic activity of Fer.

Our data indicate that Sn ions rather than F ions are responsible for increasing the peroxidase-like activity of Fer. It is possible that $Sn^{2+}$ in close proximity to the nanozyme core could efficiently accelerate the $Fe^{2+}/Fe^{3+}$ redox cycles, while Sn-bound in the vicinity could serve as electron donors, resulting in electron transfer between Sn and Fe, thereby increasing ROS production. However, further studies are required to understand the exact mechanisms for catalytic enhancement in this system. Notably, the enhancement of catalytic activity was more pronounced at acidic pH value (4.5), typically found in cariogenic biofilms, whereas minimal ROS was generated close to neutral (physiological) pH, providing high selectivity and antibiofilm activity. Taken together, it provides targeted activity under pathological conditions and operates at acidic pH values at which the anticaries action of $SnF_2$ is most effective[51].

In addition to high antibiofilm specificity and efficacy, the formation of an outer layer film containing Fe, Sn, and F can provide a 'protective shield' against enamel acid demineralization. Fluoride acts by inhibiting mineral loss at the crystal surface and enhancing the rebuilding or remineralization of calcium and phosphate in a form more resistant to subsequent acid attacks[52]. The presence of a coating of metal-rich surface precipitate or a metal-rich surface layer can make enamel more acid-resistant[53,54]. We found a film at the tooth surface that contains both Fe/Sn and F, and also variable amounts of calcium and phosphate. Calcium and phosphates contribute a protective role in preventing enamel demineralization by modulating physicochemical equilibrium and forming $CaF_2$ with fluoride that reduces acid solubility while promoting remineralization[55]. To the best of our knowledge, the formation of Fe/Sn/F polyion film has not been described previously and is potentially an innovative mechanism for caries prevention.

Despite promising results, there are some limitations, but also opportunities for further research. Although our preliminary study suggests that carboxylates play an integral role in enhancing the stability of $SnF_2$, additional analyses are needed to understand the physicochemical interactions between $SnF_2$ and Fer as well as the long-term stability of the complexes and the oxidation state of Sn in the complexes. Further studies are required to elucidate the exact mechanisms by which ROS generation is enhanced by $SnF_2$ while deeper understanding on how the metal ion-fluoride film is formed may reveal additional insights on the enamel remineralization process.

Furthermore, full toxicity studies are needed to determine the long-term effects of daily use of Fer and $SnF_2$, whereas optimization of the concentrations of Fer, $SnF_2$, and $H_2O_2$ may be required for clinical translation and product development. Nevertheless, our data reveal that Fer and $SnF_2$ potentiate the therapeutic activity through unexpected synergistic mechanisms that target both the biological (biofilm) and physicochemical (enamel demineralization) traits of dental caries simultaneously.

This simple yet effective combination therapy with fluoride co-delivery could advance current anticaries treatment while leading to the development of ROS-based modalities for other biofilm-related diseases. The search for alternative modalities encompasses unapproved compounds, where further development involves a lengthy and costly process and regulatory approval. The findings that an off-the-shelf iron oxide nanoparticle formulation has a potent topical effect at a fraction (<0.2%) of the approved systemic dosage together with low dose of $SnF_2$ that operates through complementary mechanisms of action can facilitate its path to clinical translation. This approach could be targeted for high-risk individuals prone to cariogenic biofilm accumulation without increasing the risk of fluoride overexposure. Additionally, since patients with severe childhood tooth decay are often linked with iron deficiency anemia[39,56–58], the use of Fer might have a dual benefit for these patients. The possibility that two major global health problems, i.e., tooth decay and anemia[58,59], could be treated by using Fer and $SnF_2$ opens a feasible opportunity to include the combination therapy in clinical trials for caries prevention tailored to high-risk individuals with iron-deficiency anemia.

## Methods
### In vitro biofilm model and quantitative analysis
Biofilms were formed using the saliva-coated hydroxyapatite disc (sHA) model as described elsewhere[18,33,46]. *Streptococcus mutans* (*S. mutans*) UA159 (ATCC 700610), a proven virulent and well-characterized cariogenic pathogen, was grown in ultra-filtered (10 kDa, cutoff; Millipore, Billerica, MA) tryptone-yeast extract (UFTYE) broth at 37 °C and 5% $CO_2$ to mid-exponential phase. Briefly, HA discs (surface area of $2.7 \pm 0.2\ cm^2$; Clarkson Chromatography Inc., South Williamsport, PA) were vertically suspended in 24-well plates using a custom-made wire disc holder and coated with filter-sterilized human saliva for 1 h at 37 °C. Each sHA disc was inoculated with ~$2 \times 10^5$ CFU of *S. mutans* per ml in UFTYE containing 1% sucrose (Sigma-Aldrich, ≥99.5% purity) at 37 °C and 5% $CO_2$. Topical treatment of Fer (AMAG Pharmaceuticals, Inc.) and $SnF_2$ (Sigma-Aldrich, 99% purity) or vehicle control was performed for 10 min at 0, 6, 19, and 29 h. The culture medium was changed twice daily (at 19 h and 29 h). At the end of the experimental period (43 h), the biofilms were placed in 2.8 ml of $H_2O_2$ (1%, v/v) for 5 min. After $H_2O_2$ exposure, the biofilms were removed and homogenized via bath sonication followed by probe sonication (at an output of 7 W for 30 s). The homogenized suspension was serially diluted and plated onto blood agar plates using an automated EddyJet Spiral Plater (IUL, SA, Barcelona, Spain). The number of viable cells in each biofilm were calculated by counting CFU. The remaining suspension was centrifuged at 5500 g for 10 min. Finally, the resulting cell pellets were then washed, oven-dried, and weighed. $SnF_2$ and NaF (Sigma-Aldrich, 99.99% purity) treatment groups were performed according to the same procedure in a separate experiment (Supplementary Fig. 1).

To visualize the biomass reduction and EPS degradation, SYTO 9 (485/498 nm; Molecular Probes) was used for labeling bacteria and Alexa Fluor 647-dextran conjugate (647/668 nm; Molecular Probes) was used for labeling insoluble EPS. The 3D biofilm architecture was acquired using Zeiss LSM 800 with a 20× (numerical aperture = 1.0) water immersion objective. The biofilms were sequentially scanned using diode lasers (488 and 640 nm), and the fluorescence emitted was collected with GaAsP or multialkali PMT detector (475–525 nm for

SYTO 9 and 645–680 nm for Alexa Fluor 647-dextran conjugates, respectively). ImageJ software (version 1.48) was used for biofilm visualization and quantification.

## Characterization of Fer & SnF$_2$

Fer (100 µg of Fe/ml) and Fer (100 µg of Fe/ml) + SnF$_2$ (100 µg/ml) prepared in DI water were used for determining hydrodynamic diameter and zeta potential. The measurements were carried out using a Nano-ZS 90 (Malvern Instrument, Malvern, UK) at indicated time points. TEM was performed using a Tecnai T12 (FEI Tecnai) electron microscope at 100 kV. In brief, solutions of Fer and Fer + SnF$_2$ were prepared in 0.1 M sodium acetate buffer (pH 4.5) and incubated for 1 h. After that, 5 µl of the solution of Fer or Fer + SnF$_2$ was dropped onto a TEM grid, and the liquid was dried before microscopy was conducted. $^1$H NMR spectroscopic data of CMD with or without SnF$_2$ were recorded using a Bruker DMX 500, equipped with a z-gradient amplifier and 5 mm DUAL (1H/13 C) z-gradient probe head, in D$_2$O. The absorption spectra of SnF$_2$ (250 ppm of F) were measured in 0.1 M sodium acetate buffer (pH 4.5) in the presence of various materials (1 mg/ml each) (carboxymethyl-dextran (CMD; Sigma-Aldrich), dextran (Dex, T10; Pharmacosmos, Holbaek, Denmark), citric acid (CA; Fisher Scientific, ≥99.5% purity), L-ascorbic acid (AA; Sigma-Aldrich, ≥99% purity), and poly(acrylic acid) (PAA, average molecular weight: ~2000; Sigma-Aldrich)) initially and after 24 h incubation using a Genesys 150 UV–visible spectrophotometer (Thermo Scientific, Waltham, MA). Similarly, absorption spectra of SnF$_2$ (250 ppm of F) were recorded in the presence of three concentrations of mannitol (Man; Sigma-Aldrich, ≥98% purity) (1, 2, and 10 mg/ml) under the same experimental conditions.

## ROS measurement using 3,3′,5,5′-tetramethylbenzidine (TMB) assay

The catalytic activity of Fer + SnF$_2$ was investigated by a colorimetric assay using TMB (Sigma-Aldrich, ≥99% purity) as a probe, which generates a blue color after reacting with ROS[34]. Briefly, the stock solution of TMB was made in N,N-dimethylformamide (DMF, 25 mg/ml; Sigma-Aldrich, ≥99% purity). Fer (0.5 mg of Fe/ml) and SnF$_2$ (0.5 mg/ml) were incubated (separately or combined) at room temperature in 0.1 M of sodium acetate buffer (pH 4.5) for 1 h. Afterward, 40 µl of the testing sample (Fer, SnF$_2$, or Fer + SnF$_2$) and 4 µl of TMB (100 µg) were added into 922 µl of 0.1 M sodium acetate buffer (pH 4.5), mixed by pipette and absorbance was recorded at 652 nm. Then, 34 µl of H$_2$O$_2$ (1%, v/v) was added. After 10 min additional incubation in the dark, catalytic activities were monitored at 652 nm. For the control, 40 µl of the buffer solution was taken instead of the testing sample. The effect of pH on the catalytic activity of Fer when exposed to SnF$_2$ was determined at three different pH values (4.5, 5.5, and 6.5) as described above.

The effect of sodium fluoride (NaF) (final concentration 20 µg/ml), barium fluoride (BaF$_2$) (final concentration 20 or 30 µg/ml; Sigma-Aldrich, 99.95% purity) and stannous chloride (SnCl$_2$) (final concentration 20 µg/ml; Sigma-Aldrich, ≥99.99% purity) on the catalytic activity of Fer at pH 4.5 was also investigated as described above, except after adding H$_2$O$_2$, the reaction mixture was incubated only for 5 min. In order to determine the lowest amount of SnF$_2$ required for enhancing the catalytic activity of Fer and to evaluate the effect of various amounts of SnF$_2$ (final concentration 0-80 µg/ml) on the catalytic activity of Fer (final concentration 20 µg of Fe/ml), the stock solution of the mixture of Fer and SnF$_2$ was prepared at pH 4.5 (0.1 M sodium acetate buffer) by using the predetermined amount of SnF$_2$ and then the catalytic activity was assessed at pH 4.5 using the TMB assay (5 min incubation in the presence of 1% of H$_2$O$_2$), as described above. The effect of incubation time on the catalytic activity was investigated after incubating Fer and SnF$_2$ for a predetermined time as described above (5 min incubation in the presence of 1% of H$_2$O$_2$). All

the reactions were investigated using the Genesys 150 UV–visible spectrophotometer.

## Investigation of ROS generation using o-phenylenediamine (OPD)

The enhancement of the catalytic activity of Fer in the presence of SnF$_2$ was further verified by employing OPD (Sigma-Aldrich, ≥98% purity) as a ROS tracking agent[35]. Briefly, the stock solution of the combination of Fer (0.5 mg of Fe/ml) and SnF$_2$ (0.5 mg/ml) was incubated for 1 h in 0.1 M sodium acetate buffer (pH 4.5) at room temperature. Afterward, 40 µl of the mixture of Fer (20 µg of Fe) and SnF$_2$ (20 µg) and 4 µl of OPD (100 µg) were added into 922 µl of 0.1 M sodium acetate buffer (pH 4.5) and then mixed via pipetting and absorbance was recorded at 450 nm. After adding 34 µl of H$_2$O$_2$ (1%, v/v), the mixture was further incubated for 1 min, and the absorbance was recorded at 450 nm.

## ROS study using 2′,7′-dichlorofluorescin diacetate (DCFH-DA) probe

In order to further support the enhancement of the catalytic activity of Fer in the presence of SnF$_2$, we used photoluminescence (PL) method using DCFH-DA (Sigma-Aldrich, ≥97% purity) as a ROS probing agent[36]. First, stock solutions of Fer (0.5 mg of Fe/ml) with or without SnF$_2$ (0.5 mg/ml) were incubated in 0.1 M sodium acetate buffer (pH 4.5) for 1 h at room temperature. Afterward, the working solution (final volume 2 mL) containing DCFH-DA (30 µM) and Fer (20 µg of Fe/ml) with or without SnF$_2$ (20 µg/ml) was prepared in 0.1 M sodium acetate buffer (pH 4.5). Subsequently, PL intensity was recorded at 520 nm with an excitation wavelength of 505 nm. H$_2$O$_2$ (1%, v/v) was then mixed to the reaction mixture to initiate the reaction, and the PL intensity was recorded at 520 nm at different incubation times with the excitation wavelength of 505 nm. For the control, vehicle was used.

## Comparison of hydroxyl radical (•OH) production

•OH generated by Fer and Fer + SnF$_2$ in 0.1 M sodium acetate buffer (pH 4.5) was analyzed by a PL technique using coumarin (Sigma-Aldrich, ≥99% purity) as a •OH trapping molecule[37,38]. First, stock solutions of Fer (0.5 mg of Fe/ml) with or without SnF$_2$ (0.5 mg/ml) were incubated in 0.1 M sodium acetate buffer (pH 4.5) for 1 h at room temperature. Afterward, Fer (20 µg of Fe/ml) with or without SnF$_2$ (20 µg/ml) was mixed with coumarin (0.1 mM) in a 10 mm path length cuvette, and then H$_2$O$_2$ (1%, v/v) was added to the reaction mixture to initiate the reaction. The PL intensity was recorded at 452 nm at different incubation times with an excitation wavelength of 332 nm. Vehicle was used as the control.

## Iron release study

The release of soluble iron from Fer, in the presence and absence of SnF$_2$, was investigated using inductively coupled plasma optical emission spectroscopy (ICP-OES, Spectro Genesis). Briefly, 10 ml of Fer (0.5 mg of Fe/ml) was incubated with or without SnF$_2$ (0.5 mg/ml) for 1 h in 0.1 M sodium acetate buffer (pH 4.5, 5.5, or 6.5) at room temperature. Afterward, free iron ions and intact nanoparticles were separated by centrifugation using ultrafiltration tubes (3 kDa MWCO). The pellet was then resuspended in the same volume using 0.1 M sodium acetate buffer. Subsequently, the filtrate and resuspend pellet were digested in nitric acid and finally diluted with DI water before analysis by ICP-OES.

## Stability study of SnF$_2$ after catalytic reaction

To investigate the extent of SnF$_2$ oxidation after catalytic reaction, SnF$_2$ (1 mg/ml) was mixed with Fer (1 mg of Fe/ml) in 0.1 M sodium acetate buffer (pH 4.5), and then H$_2$O$_2$ (1 %, v/v) was added to the solution to initiate the reaction. After incubating the mixture for the predetermined time with H$_2$O$_2$, the absorption spectra of the solutions were recorded following a 10-fold dilution. Furthermore, absorption spectra of SnF$_2$ (1 mg/ml) were recorded in the presence and absence

of $H_2O_2$ (1%, v/v) (10 min incubation in the presence of $H_2O_2$) in 0.1 M sodium acetate buffer (pH 4.5). Additionally, the absorption spectrum of the diluted $SnF_2$ solution was measured after 60 min incubation in the presence of $H_2O_2$.

## Determination of free tin ions after the catalytic reaction

The free tin ions from the combination of Fer + $SnF_2$, in the presence and absence of $H_2O_2$, was investigated using ICP-OES. Briefly, $SnF_2$ (1 mg/ml) was mixed with Fer (1 mg of Fe/ml) in 0.1 M sodium acetate buffer (pH 4.5) and then incubated for 24 h at room temperature. The solution was then further incubated for 10 min with or without $H_2O_2$ (1%, v/v) to initiate the catalytic reaction. Afterward, free tin ions were collected from the filtrate by centrifugation (1 h; 3184 × g) using ultrafiltration tubes (3 kDa MWCO). Subsequently, the filtrate was digested in nitric acid and diluted with DI water before analysis by ICP-OES.

## Toxicity study of the combined treatment of Fer and $SnF_2$ in immortalized human gingival keratinocytes (HGKs)

The in vitro biocompatibility of the combination of Fer and $SnF_2$ was investigated in HGK cells using an MTS [(3-(4,5- dimethylthiazol-2-yl)-5-(3-carboxymethoxyphenyl)-2-(4-sulfophenyl)-2H-tetrazolium)] assay (CellTiter 96 cell proliferation assay kit; Promega, WI, USA). HGK cells were kindly provided by the laboratory of Dana T. Graves (School of Dental Medicine, University of Pennsylvania) and were cultured in KBM-2 medium (Lonza Group AG, Basel, Switzerland). To determine the cytotoxicity, HGK cells were seeded in 96-well plates at a density of $10^4$ cells per well. Cells were then incubated at 37 °C in a humidified 5% $CO_2$ atmosphere in a cell incubator for 24 h. Afterward, old media was replaced with 100 μl of fresh media with or without Fer (1 mg of Fe/ml) and $SnF_2$ (250 ppm of F), or either alone, and incubated for 10 min. After that, the media was removed, the cells were washed twice with sterile phosphate buffered saline (PBS) and 100 μl of fresh complete cell culture media was added to each well. After 24 h incubation, the cell culture media was removed, and 20 μl of MTS reagent and 100 μl of media were added to each well. After 3 h additional incubation under standard cell culture conditions, the absorbance was recorded at 490 nm using a microplate reader. The cell viability was calculated using the following formula:

$$\text{Cell viability} = \frac{A_{490}^{treated}}{A_{490}^{untreated}} \times 100\%$$

## In vivo efficacy of Fer in combination with $SnF_2$

In vivo efficacy was assessed using a well-established rodent model of dental caries, as reported previously[44,60]. In brief, 15 days-old specific pathogen free Sprague-Dawley rat pups were purchased with their dams from Harlan Laboratories (Madison, WI, USA). Upon arrival, animals were screened for *S. mutans* by plating oral swabs on mitis salivarius agar plus bacitracin (MSB). Then, the animals were orally infected with *S. mutans* UA159, and their infections were confirmed at 21 days via oral swabbing. The treatment agents were applied on the tooth surfaces using a custom-made applicator. To simulate a clinical scenario, a topical treatment regimen was used that consisted of a short exposure (30 s) to the agent, followed by another short exposure (30 s) to $H_2O_2$ (1%, v/v) (or buffer). All infected pups were randomly placed into five treatment groups, and their teeth were treated twice daily. The treatment groups included: (1) control (0.1 M sodium acetate buffer, pH 4.5), (2) Fer only (1 mg of Fe/ml), (3) $SnF_2$ only (250 ppm of F), (4) 1/4 Fer + 1/4$SnF_2$ (0.25 mg of Fe/ml and 62.5 ppm of F), and (5) Fer + $SnF_2$ (1 mg of Fe/ml and 250 ppm of F). All the samples were prepared immediately before each treatment in 0.1 M sodium acetate buffer (pH 4.5). Each group was provided the National Institutes of Health cariogenic diet 2000 (TestDiet, St. Louis, MO) and 5% sucrose

water ad libitum. The experiment proceeded for 5 weeks, and their physical appearance was recorded daily. At the end of the experimental period, all animals were sacrificed, and their jaws were surgically removed and aseptically dissected, followed by sonication to recover total oral microbiota as reported previously[61]. All of the jaws were defleshed, and the teeth were prepared for caries scoring based on Larson's modification of Keyes' system[44]. Determination of the caries score of the jaws was performed by a calibrated examiner who was blinded for the study by using codified samples. Enamel surfaces were analyzed as described below. Moreover, the gingival tissues were collected for hematoxylin and eosin (H&E) staining for histopathological analysis by an oral pathologist at Penn Oral Pathology. This research was reviewed and approved by the University of Pennsylvania Institutional Animal Care and Use Committee (IACUC #805529).

## Preparation of enamel samples for scanning transmission electron microscopy (STEM)

We identified the most promising location for focused ion beam (FIB) lift-out as the middle cusp of the buccal side by assessing curvature and roughness using synchrotron micro-computed tomography reconstructions of whole molars (mandibular) and 3D measuring laser confocal microscopy (Olympus LEXT OLS5000 equipped with a laser operating at a wavelength of 405 nm). Whole air-dried M1 molars were attached, with the buccal side facing up, to a scanning electron microscopy (SEM) stub with carbon and copper tape (Electron Microscopy Sciences). Specimens were then coated with carbon (~10 nm, Denton Desk deposition system). The surface of the middle cusp of the buccal side of the tooth was then investigated in detail for microscopic surface roughness, using electron beam imaging at a high tilt angle (52°). Lamellae were lifted out directly from the surface of the tooth in areas that were sufficiently flat (≲500 nm height modulation), using a dual-beam FIB/SEM (FEI Helios Nanolab 600 FIB/SEM) with a gallium liquid metal source ion source (LMIS) operated at an accelerating voltage of 5–30 kV. Initially, a ~100 nm layer of protective carbon was deposited using the electron beam (5 kV, 1.4 nA) on a 2 μm × 10 μm area of interest using a gas injection system (GIS) through decomposition of a phenanthrene precursor gas. A ~1 μm protective platinum layer was then deposited on top of the carbon using the ion beam (30 kV, 93 pA) through decomposition of a (methylcyclopentadienyl)-trimethyl platinum precursor gas. Next, two trenches were cut (30 kV, 6.5 nA) and edged-cleaned at slightly lower currents (30 kV, 2.8 nA) to allow for a roughly 1.5 μm thick lamella. Following an in situ lift-out procedure, a tungsten micromanipulator (Oxford Instruments) was then welded onto the lamella using platinum, and the sample was cut loose from the bulk material. After mounting the lamella as a flag onto one of the four posts of a TEM Cu half-grid (Ted Pella), the lamella was thinned in a 5 μm wide window (5 kV, 81 pA) and cleaned at low voltage and current (2 kV, 28 pA) until a final thickness of roughly 20–80 nm was achieved near the surface of the lamella.

## Scanning transmission electron microscopy (STEM) with energy dispersive spectroscopy (STEM-EDS) and electron energy loss spectroscopy (STEM-EELS)

Imaging of enamel specimens was performed using an JEOL GrandARM 300 F with a cold-cathode field-emission electron gun used at an accelerating voltage of 300 kV, using a probe current of ~204 pA with a dwell time of 10 μs. The collection semi-angle used was 106–180 mrad for high-angle annular dark-field (HAADF) imaging. Elemental maps were recorded using EDS using a windowless 100 mm² Xmax$^N$ 100TLE Silicon Drift detector (SDD) with a solid angle of approximately 0.98 sr (Oxford Instruments NanoAnalysis) with a resolution of 1024 × 1024 pixels with a dwell time of 10 μs per pixel. Elemental maps were binned (4 × 4) and converted to mole fractions, using QuantMap (AZtecTEM). Binned mole fraction maps were then exported for further processing and visualization using Matlab 2022b (Mathworks, Natick, MA).

Line profiles (mole fractions as a function of distance in the direction normal to the external enamel surface) were determined by resampling regions of interest (ROIs) within elemental maps (determined by EDS) on a rectangular query grid rotated such that the y-direction was normal to the interface, as assessed from Ca maps. Resampling by linear interpolation was carried out using the griddedInterpolant() function included in Matlab r2022b (Mathworks, Natick, MA). Resampled ROIs were then averaged in the direction parallel to the interface. The position of the outer surface in treated and untreated samples, of the interface between the Fe and Sn rich layer and underlying enamel were identified manually from line profiles. Profiles were aligned on the outer surface position, and the distance axis was set to zero at the interface between the Fe and Sn rich layer and enamel. Data were plotted as the mean value at the given distance (solid circles), and in smoothed form (lines), as the local 3-point mean (moving average with span 3, using the movmean() function).

EEL spectra were acquired with a GIF continuum system (Gatan) using a K3 IS direct electron detector (Gatan) in counting mode at 300 kV. The high quantum efficiency of this detector (DQE up to 90%) allowed the simultaneous acquisition of the relevant inner shell ionization (core loss) edges and zero loss region at high energy resolution, except for the phosphorous K and L edges, which were outside the selected energy range. The convergence semi-angle of the probe was 19 mrad, and the probe current was ~27 pA, as determined using a Faraday cup. The collection semi-angle of 36 mrad was defined by the EELS entrance aperture (5 mm). The three-dimensional spectrum image dataset was collected using an energy dispersion of 0.35 eV/channel and the probe dwell time was 4 ms/pixel with a pixel size of 6 nm, with sub-pixel scanning enabled (32 × 32) to yield a ~3.8 Å pixel. Simultaneously, ADF images were acquired using a collection semi-angle of 51–115 mrad. In post-processing, the zero-loss peak was aligned in every pixel of the spectrum image using GMS software (Gatan, Inc). Elemental Quantification Analysis was performed in the same software, using a Hartree-Slater cross-section model and including plural scattering corrections.

### High-resolution TEM (HRTEM) imaging
HRTEM imaging of enamel specimens was performed using an JEOL GrandARM 300F at an accelerating voltage of 300 kV. Images (edge length: 4096 pixels, scale factor 0.0328 nm/pixel) were processed using Matlab r2022b (Mathworks, Natick, MA). Two-dimensional Fourier transforms of regions of interest (edge length: 1024 pixels) were determined using fft2() and rearranged using fftshift() to move the zero frequency components to the center of the image. Fourier transform images were unwrapped in the azimuthal direction by interpolation using griddedInterpolant() with a query grid in polar coordinates (radial pitch: 0.0298 nm$^{-1}$/pixel; azimuthal pitch: 1°/pixel) and integrated in the azimuthal direction.

### X-ray photoelectron spectroscopy (XPS)
Two mandibular (M1) rat molars, one from Fer + SnF$_2$ treated group and one from control group, were dissected and attached using copper tape (Electron Microscopy Sciences). XPS analysis was conducted using a Thermo Scientific Nexsa G2 using an Al-Ka X-ray source, with the following parameters: pressure of 2·10$^{-9}$ torr (2.5·10$^{-7}$ Pa), an X-ray gun power of 150 W, a spot diameter of 100 μm, and a takeoff angle of 0°. XPS survey spectra were acquired under a pass energy of 100 eV, using a step size of 1 eV. High-resolution spectra for F, Fe, Ca, P, O, and Sn were acquired under a pass energy of 50 eV, using a step size of 0.1 eV, and averaging over 10 scans. For depth profiling, the surface was excavated using an argon ion beam (4 keV, diameter 500 μm, 'high current' mode, 30–300 s increment) between successive spectra. All data were processed using Avantage (Thermo Scientific), and spectra were referenced to adventitious carbon at 284.8 eV.

### 16S rRNA sequencing
Cells were pelleted from dental plaque by centrifuging at maximum speed for 5 min. DNA was extracted from the pellets using the Qiagen DNeasy PowerSoil htp kit according to the manufacturer's instructions within a sterile class II laminar flow hood. Mock washes and mock extractions were included as control for microbial DNA contamination arising through the sonication and extraction processes, respectively.

Polymerase chain reaction (PCR) amplification of V1-V2 region of 16S rRNA gene was performed using Golay-barcoded universal primers 27F and 338R. Four replicate PCR reactions were performed for each sample using Q5 Hot Start High Fidelity DNA Polymerase (New England BioLabs). Each PCR reaction contained: 4.3 μl microbial DNA-free water, 5 μl 5X buffer, 0.5 μl dNTPs (10 mM), 0.17 μl Q5 Hot Start Polymerase, 6.25 μl each primer (2 μM), and 2.5 μl DNA. PCR reactions with no added template or synthetic DNAs were performed as negative and positive controls, respectively[62]. PCR amplification was done on a Mastercycler Nexus Gradient (Eppendorf) using the following conditions: DNA denaturation at 98 °C for 1 min, then 20 cycles of denaturation at 98 °C for 10 s, annealing 56 °C for 20 s and extension 72 °C for 20 s, last extension was at 72 °C for 8 min. PCR replicates were pooled and then purified using a 1:1 ratio of Agencourt AMPure XP beads (Beckman Coulter, Indianapolis, IN), following the manufacturer's protocol. The final library was prepared by pooling 10 μg of amplified DNA per sample. Those that did not arrive at the DNA concentration threshold (e.g., negative control samples) were incorporated into the final pool by adding 12 μl. The library was sequenced to obtain 2 × 250 bp paired-end reads using the MiSeq Illumina[63].

To analyze 16S rRNA gene sequences, we used QIIME2 v19.4[64]. We obtained taxonomic assignments based on GreenGenes 16S rRNA database v.13_8[65] and ASV analysis of shared and unique bacterial taxa through DADA2[66]. PCoA was performed using library ape for R programming language[67]. To test the differences between communities, we used library vegan and UniFrac distances (https://CRAN.R-project.org/package=vegan). R environment (version 4.0.3) was used for statistical analysis. Non-parametrical test Wilcoxon Rank Sum Test was performed for the pairwise comparison between treatment groups for richness and Shannon diversity analysis. PERMANOVA analysis was performed for weighted UniFrac principal coordinate analysis to evaluate the differences between treatment groups. Statistical significance was considered <0.05.

### Statistical analysis
The data presented as the mean ± standard deviation were performed at least three times independently unless otherwise stated. One-way analysis of variance (ANOVA) followed by the Tukey test was used to determine the statistical significance between the control and the experimental groups unless otherwise stated. *p* values < 0.05 were considered statistically significant.

### Reporting summary
Further information on research design is available in the Nature Portfolio Reporting Summary linked to this article.

## Data availability
16S rRNA sequencing data is available in the public repository NCBI under the accession number PRJNA914620. All the other data that support the findings of this study are available in the main text or the supplementary information. Source data are provided with this paper.

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

## Acknowledgements

This work was supported by the NIH grant RO1-DE025848 to H.K. This work made use of the EPIC facility of Northwestern University's NU*ANCE* Center, which has received support from the SHyNE Resource (NSF ECCS-2025633), the IIN, and Northwestern's MRSEC program (NSF DMR-1720139). Research reported in this publication was supported in part by instrumentation provided by the Office of the Director, National Institutes of Health, under Award Number S10OD026871. S.B. was in part supported by the National Institutes of Health's National Center for Advancing Translational Sciences, Grant Number TL1TR001423. The content is solely the responsibility of the authors and does not necessarily represent the official views of the National Institutes of Health. Z.R. is supported by the National Institute of Dental and Craniofacial Research Postdoctoral Training Program under Award R90DE031532. We would also like to thank Liam Spillane from Gatan, Inc. for aid with the STEM-EELS acquisition. A portion of the schematic diagram in Fig. 1e was generated using BioRender.com (Agreement number: UO25QG4RN3).

## Author contributions

Y.H., Y.Liu, and N.K.P. contributed equally. Y.H., Y.Liu, N.K.P., D.P.C., and H.K. conceived and designed the experiments. Y.H., Y.Liu, N.K.P., D.P.C., and H.K. wrote the manuscript. Y.H., Y.Liu., N.K.P., S.S., A.S., J.C.H., Z.R., Z.X., D.K., T.I., M.J.O., and Y.Li collected and analyzed the data. F.A. evaluated tissue slices for the in vivo study. P.J.M.S. S.B., X.Z., and D.J. designed and performed the experiments on the structure and composition of enamel at the surface of rat molars from the in vivo study. C.B. and D.T.Z. provided suggestions and technical support on the project. D.P.C. and H.K. conceptualized the manuscript. All authors discussed the results, critically revised the manuscript, and gave the final approval.

## Competing interests

Y.H., H.K., D.P.C., and N.K.P. are listed as inventors on a patent related to the subject matter of this work (U.S. Provisional Application No.: 63/414,757). All other authors declare no competing interests.

## Additional information

[1]Department of Radiology, Perelman School of Medicine, University of Pennsylvania, Philadelphia, PA, USA. [2]Biofilm Research Labs, Levy Center for Oral Health, School of Dental Medicine, University of Pennsylvania, Philadelphia, PA, USA. [3]Department of Orthodontics and Divisions of Pediatric Dentistry and Community Oral Health, School of Dental Medicine, University of Pennsylvania, Philadelphia, PA, USA. [4]Department of Preventive and Restorative Sciences, School of Dental Medicine, University of Pennsylvania, Philadelphia, PA, USA. [5]Department of Bioengineering, School of Engineering and Applied Sciences, University of Pennsylvania, Philadelphia, PA, USA. [6]Department of Stomatology, Dental School, University of Seville, Seville, Spain. [7]Center for Innovation and Precision Dentistry, School of Dental Medicine, School of Engineering and Applied Sciences, University of Pennsylvania, Philadelphia, PA, USA. [8]Department of Preventive Dentistry, School of Dentistry, Jeonbuk National University, Jeonju, Republic of Korea. [9]Department of Pediatric Dentistry, Nihon University School of Dentistry at Matsudo, Chiba, Japan. [10]Department of Chemical and Biomolecular Engineering, University of Pennsylvania, Philadelphia, PA, USA. [11]Department of Cariology, Operative Dentistry and Dental Public Health and Oral Health Research Institute, Indiana University School of Dentistry, Indianapolis, IN, USA. [12]Department of Basic and Translational Sciences, School of Dental Medicine, University of Pennsylvania, Philadelphia, PA, USA. [13]Northwestern University Atomic and Nanoscale Characterization Experimental Center, Northwestern University, Evanston, IL, USA. [14]Department of Materials Science and Engineering, Northwestern University, Evanston, IL, USA. [15]These authors contributed equally: Yue Huang, Yuan Liu, Nil Kanatha Pandey.
✉e-mail: david.cormode@pennmedicine.upenn.edu; koohy@upenn.edu

