## [Peer Review File · Nature Communications]

Reviewers' Comments:

Reviewer #1:

Remarks to the Author:

This is an interesting manuscript describing the improved effect of iron oxide nanozymes when used in association with stannous fluoride. This is of high relevance, as the iron oxide nanozymes have been explored in detail as an anticaries platform by this research group, and the addition of fluoride can further improve the anticaries effect. Surprisingly, as noted by the authors, the combination does not only have additive effects, but synergistic effects. Because ferumoxytol is cleared by the FDA for systemic treatment of iron deficiency, the newly investigated combination has an impactful, translational potential to address two highly prevalent diseases of public health concern, iron deficiency anemia and severe childhood caries.

The experiments described are all of high quality and aim for an in-depth investigation of many of the mechanisms observed. This review points out some suggestions for the improvement of the writing and to make this contribution even more relevant in terms of reproducibility.

1. Abstract:

Lines 42-43: "fluoride, the mainstay anticaries (tooth-enamel protective) agent". I believe the authors are trying to highlight that the main effect of fluoride does not rely on its antibiofilm activity, but on its chemical effect on de-remineralization processes. That could be made clearer, perhaps, by removing the "tooth-enamel protective" (which may sound as if the incorporation of fluoride in the enamel is the main mechanism of its "protective" effect, which is not entirely true), and adding, at the end of the sentence, "that has negligible effect on the biofilm", or "that acts mainly through a chemical effect on de-remineralization".

2. Introduction:

a. Lines 78-80: although the sentence on fluoride toxicity is not totally untrue, it can be argued that current methods of fluoride use (at currently used doses, concentrations) do work while posing no risks of dental fluorosis of aesthetic or public health concern. So I suggest the authors rephrase this sentence, especially the first part.

b. Line 107: Perhaps adding "caries" before biological would improve the understanding. "... that target the caries biological (biofilm) and physicochemical (enamel demineralization) traits..."

3. Results:

a. Figure 5, legend: in the last sentence (line 365), there seems to be an extra "that". Please revise.

b. Figure 5C: Please see the comment below for supplementary material for the description of "smoothed".

4. Discussion:

Lines 521-522: Please check the sentence starting with "It is noteworthy...", it can be rephrased improving the understanding.

5. Methods:

a. The authors describe that all data result from at least 3 independent repetitions, but in none of the figures the n values are presented. Standard deviations are dependent on the n, so it would be relevant to have that information added to the legend figures or methods. For example, I believe that more than 3 animals were used in each of the groups for the in vivo study, and adding that "n" would increase the understanding of the variability for the different outcomes.

b. The authors tested for the stability of the treatments over time, especially for the combination of Fer+SnF₂ (ex., fig. 3). It would be relevant to add information about when the treatments were prepared before each experiment. For example, for the rat study, were the treatments prepared right before each application? Or every day? Or every week?

c. Lines 544-545: It sounds confusing to describe the SnF₂ and NaF groups at the end of this sentence. Were they tested in a different experiment?

d. Lines 581-588: It seems that this paragraph is mainly a repetition of the previous one. Can't they be combined?

e. Lines 664-667: How were the treatments applied to the rats' teeth? Was it an active application (ex., with a brush) for 30 s? Or were the treatments delivered (ex., with a pipette or applicator) slowly, for about 30 s?

6. Supplementary information:

Figure S12: The legend says that the graphs depict information from "rats" (please check spelling vs "rates") treated with Fer+SnF2 and untreated controls. Which graphs represent treated or untreated? Also, is "smoothed" the integration of the "treated" dots? If so, the use of "treated" for the dots and "smoothed" for the integration seems confusing, in case both refer to the "treated" samples. Please check/clarify. (This same comment applies to Fig 5F.)

Reviewer #2:

Remarks to the Author:

In this paper, the authors found that synergistic effects have been achieved when Fer nanozyme combined with SnF2 to inhibit biofilm accumulation and enamel damage. On one hand, SnF2 enhanced the catalytic activity of Fer, increasing antibiofilm efficiency. On the other hand, Fer enhanced the stability of SnF2 in aqueous solution. Furthermore, the authors revealed that Sn2+ bound to carboxylate groups in Fer is the key way to stabilize SnF2 and boost catalytic activity. Importantly, Fer in combination with SnF2 demonstrated high efficacy for dental caries treatment, preventing enamel demineralization and cavitation altogether without adverse effects on the host tissues or causing changes in the oral microbiome diversity. The work will extend the translational application of iron oxide nanozyme in oral health. I thus recommend it to be accepted for publication on Nature Communications, after addressing below questions with minor revision.

1. What is the ratio of SnF2 to enhance the catalytic activity of Fer? It seems that SnF2 increased the catalytic efficiency dramatically. Is it possible to measure or estimate the number of SnF2 on each Fer nanozyme?

2. Is the binding of SnF2 on Fer reversible? In particular, when conducting catalytic reaction, will SnF2 retain on Fer? Does the binding have pH dependence? It was shown that the binding occurred at pH 4.5. How about other pH?

3. For all the tests, such as antibiofilm, catalytic activity, in vivo caries treatment, H2O2 was used at 1% concentration. Please clarify the reason to use such specific concentration. Is it possible to use less level of H2O2?

4. In line 192-193, it is mentioned that "consistent with dynamic light scattering (DLS) data, mixing Fer with Sn F2 did not seem to affect the size of Fer. However, there is no data for DLS in the manuscript.

Reviewer #3:

Remarks to the Author:

The authors describe a study to examine the synergistic effect between the approved agents stannous fluoride and Ferumoxytol (aqueous iron oxide nanoparticle) to target biofilms associated with dental caries and to prevent enamel demineralisation. The results achieved are particularly significant within the field and compare very favourably with the established literature. This is a very well conducted study that examines the biological, chemical, and physical mechanisms involved. The work certainly supports the conclusions and claims made.

Attention should be paid to the following:

Intro / line 77 – reference should be made to salivary buffering capacity and how this can differ between individuals.

Results section needs to be carefully checked as regards correct use of English language.

Results / line 335 – *S. mutans* – please state designation of strain – assume this is of human origin.

Results / line 415 – beneficial species were increased. Further discussion need as regards properties e.g., nitrate-reducing capacity / interaction with pathogens.

Results / line 432 – please comment on increases in richness / diversity. Only 'main bacterial genera' shown in fig. 6D – please comment on any significant changes at the species level. Were populations of *S. mutans* monitored in the mouse model?

Note: All additions/changes are highlighted in blue color in the revised manuscript, but not in the revised supplementary information file, since the formatting instructions say, “Remove track changes and highlights” from the revised supplementary information file.

REVIEWER COMMENTS

Reviewer #1 (Remarks to the Author):

This is an interesting manuscript describing the improved effect of iron oxide nanozymes when used in association with stannous fluoride. This is of high relevance, as the iron oxide nanozymes have been explored in detail as an anticaries platform by this research group, and the addition of fluoride can further improve the anticaries effect. Surprisingly, as noted by the authors, the combination does not only have additive effects, but synergistic effects. Because ferumoxytol is cleared by the FDA for systemic treatment of iron deficiency, the newly investigated combination has an impactful, translational potential to address two highly prevalent diseases of public health concern, iron deficiency anemia and severe childhood caries.

The experiments described are all of high quality and aim for an in-depth investigation of many of the mechanisms observed. This review points out some suggestions for the improvement of the writing and to make this contribution even more relevant in terms of reproducibility.

We thank the reviewer for the positive comments. We have addressed each of the comments below and made revisions as suggested.

1. Abstract:

Lines 42-43: "fluoride, the mainstay anticaries (tooth-enamel protective) agent". I believe the authors are trying to highlight that the main effect of fluoride does not rely on its antibiofilm activity, but on its chemical effect on de-remineralization processes. That could be made clearer, perhaps, by removing the "tooth-enamel protective" (which may sound as if the incorporation of fluoride in the enamel is the main mechanism of its "protective" effect, which is not entirely true), and adding, at the end of the sentence, "that has negligible effect on the biofilm", or "that acts mainly through a chemical effect on de-remineralization".

Response: Thank you for this valuable comment. In order to avoid confusion, we have deleted “tooth-enamel protective” in the revised manuscript and changed the sentence so that it now reads (please see first paragraph in page 2 in the revised manuscript):

“Dental caries (tooth decay) is the most prevalent human disease caused by oral biofilms, affecting nearly half of the global population despite increased use of fluoride, the mainstay anticaries agent, which protects enamel against acid damage but has limited antibiofilm effect.”

2. Introduction:

a. Lines 78-80: although the sentence on fluoride toxicity is not totally untrue, it can be argued that current methods of fluoride use (at currently used doses, concentrations) do work while posing no risks of dental fluorosis of aesthetic or public health concern. So I suggest the authors rephrase this sentence, especially the first part.

Response: As the reviewer suggested, we revised the sentence as follows (see page 3, first paragraph, in the revised manuscript):

“Although currently used fluoride doses provide minimal toxicity, there are potential risks associated with fluoride overexposure (e.g., dental fluorosis), especially for young children.”

b. Line 107: Perhaps adding "caries" before biological would improve the understanding. "... that target the caries biological (biofilm) and physicochemical (enamel demineralization) traits..."

Response: We have edited as follows (see page 4, last paragraph, in the revised manuscript):

"... that target the biological (biofilm) and physicochemical (enamel demineralization) traits of dental caries while..."

3. Results:

a. Figure 5, legend: in the last sentence (line 365), there seems to be an extra "that". Please revise.

Response: We made this change.

b. Figure 5C: Please see the comment below for supplementary material for the description of "smoothed".

Response: We believe the reviewer is referring to Fig. 5f. Please see our response in comment no.

6.

4. Discussion:

Lines 521-522: Please check the sentence starting with "It is noteworthy...", it can be rephrased improving the understanding.

Response: We have revised the sentence to read:

“Additionally, since patients with severe childhood tooth decay are often linked with iron deficiency anemia, the use of Fer might have a dual benefit for these patients.” (see page 27, above the section ‘Methods’ in the revised manuscript)

5. Methods:

a. The authors describe that all data result from at least 3 independent repetitions, but in none of the figures the n values are presented. Standard deviations are dependent on the n, so it would be relevant to have that information added to the legend figures or methods. For example, I believe that more than 3 animals were used in each of the groups for the in vivo study, and adding that "n" would increase the understanding of the variability for the different outcomes.

Response: Thank you for this important comment. We have added “n” values for each experiment in the revised manuscript and in the revised supplementary information file.

b. The authors tested for the stability of the treatments over time, especially for the combination of Fer+SnF₂ (ex., fig. 3). It would be relevant to add information about when the treatments were prepared before each experiment. For example, for the rat study, were the treatments prepared right before each application? Or every day? Or every week?

Response: We apologize for our lack of clarity on this point. For all experiments, unless noted otherwise, all samples, including Fer+SnF₂, were prepared immediately before each experiment. To address this point, we have added this information in the method section (page 34, in the revised manuscript).

“All the samples were prepared immediately before each treatment in 0.1 M sodium acetate buffer (pH 4.5).”

c. Lines 544-545: It sounds confusing to describe the SnF₂ and NaF groups at the end of this sentence. Were they tested in a different experiment?

Response: We agree. The SnF₂ and NaF were tested in a different experiment. We have revised the description to make it clear (please see page 28 in the revised manuscript).

“SnF₂ and NaF treatment groups were performed according to the same procedure in a separate experiment (Supplementary Fig. 1)”.

d. Lines 581-588: It seems that this paragraph is mainly a repetition of the previous one. Can't they be combined?

Response: We agree with the reviewer. So, we have merged them (page 30, first paragraph in the revised manuscript), so that it now reads:

“The effect of sodium fluoride (NaF) (final concentration 20 µg/ml, Sigma-Aldrich), barium fluoride (BaF₂) (final concentration 20 or 30 µg/ml, Sigma-Aldrich) and stannous chloride (SnCl₂) (final concentration 20 µg/ml, Sigma-Aldrich) on the catalytic activity of Fer at pH 4.5 was also investigated as described above, except after adding H₂O₂, the reaction mixture was incubated only for 5 min.”

e. Lines 664-667: How were the treatments applied to the rats' teeth? Was it an active application (ex., with a brush) for 30 s? Or were the treatments delivered (ex., with a pipette or applicator) slowly, for about 30 s?

Response: The treatment agents were applied on the tooth surfaces using a custom-made applicator and the solution was kept in the mouth for 30 s after topical application to simulate the clinical situation. We have edited the methods section to clarify this point (see page 33 in the revised manuscript), so that it now reads:

“The treatment agents were applied on the tooth surfaces using a custom-made applicator.”

6. Supplementary information:

Figure S12: The legend says that the graphs depict information from "rats" (please check spelling vs "rates") treated with Fer+SnF₂ and untreated controls. Which graphs represent treated or untreated? Also, is "smoothed" the integration of the "treated" dots? If so, the use of "treated" for the dots and "smoothed" for the integration seems confusing, in case both refer to the "treated" samples. Please check/clarify. (This same comment applies to Fig 5F.)

Response: Thank you for noticing this typo. We corrected the spelling. The figure only shows the treated group. Please see the updated captions. We also reiterate key points in the new captions (Supplementary Fig. 13 and Fig. 5f) to clarify the meaning of “smoothed”; see below (Figure S12 is now Figure S13):

“Supplementary Fig. 13. Elemental composition of the surface of a rat M1 molar treated with Fer+SnF₂ as assessed by STEM-EELS. **a-f** Plot of the mean mole fraction of Ca (blue line), and of the sum of the mean mole fractions of Fe, Sn, and F (red line, **a**), and plot of mean mole fractions of Ca (**b**), O (**c**), F (**d**), Fe (**e**), and Sn (**f**) vs. distance in the direction normal to the EES. The distance axis is referenced to the approximate position of the interface between the Fe/Sn/F-rich layer and the underlying enamel. For **b-f**, solid circles indicate the mean mole fraction at a given distance, and lines indicate the moving average of the mole fraction with span 3 (denoted as “smoothed” in legend). Note that the data shown here was generated from a separately prepared sample extracted from the same tooth shown in Fig. 5f.”

“Fig. 5f. Elemental composition of the surface of M1 molars of rats treated with Fer+SnF₂, and from untreated controls, was assessed by STEM-EDS. Plot of the sum of mean mole fractions for Ca and P, and for Fe, Sn, and F (**i**), and plots of mean mole fraction for Ca (**ii**), O (**iii**), F (**iv**), Fe (**v**), and Sn (**vi**) vs. distance in the direction normal to the EES. Profiles were manually aligned on the outer surface, and that the distance axis is referenced to the approximate position of the interface between the Fe/Sn/F-rich layer and the underlying enamel of the treated sample. For (**ii**)-(**vi**), solid circles indicate the mean mole fraction at a given distance, and lines indicate the moving average of the mole fraction with span 3 (denoted as “smoothed” in legend).”

Reviewer #2 (Remarks to the Author):

In this paper, the authors found that synergistic effects have been achieved when Fer nanozyme combined with SnF₂ to inhibit biofilm accumulation and enamel damage. On one hand, SnF₂ enhanced the catalytic activity of Fer, increasing antibiofilm efficiency. On the other hand, Fer enhanced the stability of SnF₂ in aqueous solution. Furthermore, the authors revealed that Sn²⁺ bound to carboxylate groups in Fer is the key way to stabilize SnF₂ and boost catalytic activity. Importantly, Fer in combination with SnF₂ demonstrated high efficacy for dental caries treatment, preventing enamel demineralization and cavitation altogether without adverse effects on the host tissues or causing changes in the oral microbiome diversity. The work will extend the translational application of iron oxide nanozyme in oral health. I thus recommend it to be accepted for publication on Nature Communications, after addressing below questions with minor revision.

We thank the reviewer for the positive comments and recommendation of publication. We have addressed each of the comments and made revisions as suggested.

1. What is the ratio of SnF₂ to enhance the catalytic activity of Fer? It seems that SnF₂ increased the catalytic efficiency dramatically. Is it possible to measure or estimate the number of SnF₂ on each Fer nanozyme?

Response: Thanks for this comment, and we agree that SnF₂ increases the catalytic efficiency of Fer drastically. We performed an additional experiment to determine the lowest amount of SnF₂ that can enhance the catalytic activity of Fer. As depicted in Supplementary Fig. 6, even a very small amount of SnF₂ (0.156 µg/ml) can enhance the catalytic activity of Fer (20 µg of Fe/ml). Therefore, the ratio of the mass of SnF₂ to the mass of iron (Fe) to enhance the catalytic activity is determined to be 0.0078:1. (Since the catalytic activity of Fer arises from 'Fe', we use the mass of 'Fe' rather than the mass of 'Fer'.) We have discussed it in the revised manuscript (page 14):

“Surprisingly, a very small amount of SnF₂ (0.156 µg/ml) is sufficient for enhancing the catalytic activity of Fer (Supplementary Fig. 6), and the ratio of the mass of SnF₂ to the mass of Fe is determined to be 0.0078:1 (under the present experimental conditions).”

Supplementary Fig. 6. Effect of 0.039-0.625 µg/ml (a) and 1.25-80 µg/ml (b) of SnF₂ on the catalytic activity of Fer (20 µg of Fe/ml) in 0.1 M sodium acetate buffer (pH 4.5), as assessed by TMB colorimetric assay. 0 µg/ml indicates Fer alone. The data are presented as mean ± standard deviation (n=3). **p* < 0.05, ***p* < 0.01, ****p* < 0.001; ns, nonsignificant; one-way ANOVA followed by Tukey test.

We have also included this point in the method section of the revised manuscript (page 30, first paragraph):

“In order to determine the lowest amount of SnF₂ required for enhancing the catalytic activity of Fer and to evaluate the effect of various amounts of SnF₂ (final concentration 0-80 μg/ml) on the catalytic activity of Fer (final concentration 20 μg of Fe/ml), the stock solution of the mixture of Fer and SnF₂ was prepared at pH 4.5 (0.1 M sodium acetate buffer) by using the predetermined amount of SnF₂ and then the catalytic activity was assessed at pH 4.5 using the TMB assay (5 min incubation in the presence of 1% of H₂O₂), as described above.”

Is it possible to measure or estimate the number of SnF₂ on each Fer nanozyme?

This is an interesting question, which would require in-depth and detailed physicochemical analyses for precise measurements. We tried to estimate the theoretical number of SnF₂ on each Fer when the ratio of the mass of SnF₂ to the mass of the core of Fer (i.e., Fe₃O₄) is 1:1. Although the binding of SnF₂ with Fer occurs through tin-carboxylate complexation, the following calculation assumes that the same number of SnF₂ binds with the core of Fer (i.e., Fe₃O₄) and Fer itself.

Nanozyme core diameter (d) = 7.15 nm (this is average diameter) ¹

$$\text{Volume of Fe}_3\text{O}_4 (V) = \frac{4}{3}\pi r^3 = 191.39 \text{ nm}^3$$

Density of Fe₃O₄ (D) = 5.24 g/cm³ (assuming that the density of a Fe₃O₄ is the same as the bulk density)

$$\text{Mass of Fe}_3\text{O}_4 (M) = D \times V = 10^{-18} \text{ g} = 10^{-15} \text{ mg}$$

$$\text{Number of Fe}_3\text{O}_4 \text{ in } 1 \text{ mg/ml} = (\text{Total mass in } 1 \text{ ml})/(\text{Mass of a single Fe}_3\text{O}_4) = 10^{15}$$

$$\text{Formula weight of SnF}_2 = 156.71 \text{ g/mol} = 156710 \text{ mg/mol}$$

$$\text{Number of moles of SnF}_2 \text{ in } 1 \text{ mg} = 1/156710 = 6.381 \times 10^{-6}$$

$$\text{Number of Sn}^{2+} \text{ ions in } 1 \text{ mg} = 6.022 \times 10^{23} \times 6.381 \times 10^{-6} = 3.843 \times 10^{18}$$

$$\text{Number of Sn}^{+2} \text{ ions to each Fe}_3\text{O}_4 \text{ nanoparticle} = (3.843 \times 10^{18})/10^{15} = 3843$$

The above calculation shows that number of Sn^{+2} ions to each Fer is approximately 3843. Since this calculation is based on approximations and estimations, further experimental work is needed to determine the exact number of SnF_2 to each Fer. Since binding of SnF_2 with Fer depends upon various factors including pH, incubation time, and concentration, we believe that the number of SnF_2 that interact with Fer vary with reaction conditions. Thus, further in-depth and detailed physicochemical and biological analyses are required for precise measurements, which can be an important direction for future studies and clinical translation of Fer/ SnF_2 formulations.

2. Is the binding of SnF_2 on Fer reversible? In particular, when conducting catalytic reaction, will SnF_2 retain on Fer? Does the binding have pH dependence? It was shown that the binding occurred at pH 4.5. How about other pH?

Response: Thanks for this valuable suggestion. As SnF_2 behaves differently with different pH values, the interactions of SnF_2 and Fer can be pH dependent. To evaluate this possibility, we investigated whether Fer and SnF_2 incubated at different pH values (4.5, 5.5, and 6.5) have the same catalytic activity. First, we incubated the mixture of Fer and SnF_2 at pH 4.5, 5.5, and 6.5 for 1 h, and then centrifuged the mixtures 3 times with respective buffers using ultrafiltration tubes (3 kDa MWCO). Subsequently, pellets of Fer+ SnF_2 were dispersed in a volume equal to the volume of filtrate, and the catalytic activities of the dispersed pellets were compared using the TMB assay. As can be seen in Supplementary Fig. 5, the Fer+ SnF_2 obtained at pH 4.5 exhibited higher catalytic activity than the Fer+ SnF_2 from pH 5.5 or 6.5 and Fer alone, suggesting that the highest binding interaction of SnF_2 with Fer occurs at pH 4.5. However, further studies are needed to assess whether SnF_2 retain on Fer after catalytic reaction or whether the binding of SnF_2 is reversible. We have added the following information in the revised manuscript (page 13, last paragraph):

“Given the pH-dependent activity, we also tested binding interactions between SnF_2 and Fer at different pH values; our preliminary finding showed that the highest co-binding occurred at pH 4.5 (Supplementary Fig. 5). However, further studies are needed to assess whether SnF_2 retains on Fer after catalytic reaction or whether the binding of SnF_2 is reversible at different pH values.”

Supplementary Fig. 5. Comparison of catalytic activities of Fer+SnF₂ prepared at different pH values (4.5, 5.5, and 6.5; 0.1 M sodium acetate buffer). Briefly, Fer was incubated with SnF₂ for 1 h at three different pH (4.5, 5.5, and 6.5) and washed three times with respective buffers using ultrafiltration tubes (3 kDa MWCO). Subsequently, the pellets were resuspended in a volume equal to the volume of the filtrate. All the catalytic activities were then determined (5 min incubation in the presence of 1% of H₂O₂) using TMB assay in 0.1 M sodium acetate buffer at pH 4.5. For the purpose of comparison, Fer alone (incubated at pH 4.5) was treated in a similar way. The data are presented as mean ± standard deviation (n=3). ***p* < 0.01, ****p* < 0.001; ns, nonsignificant; one-way ANOVA followed by Tukey test.

3. For all the tests, such as antibiofilm, catalytic activity, in vivo caries treatment, H₂O₂ was used at 1% concentration. Please clarify the reason to use such specific concentration. Is it possible to use less level of H₂O₂?

Response: We selected this concentration based on our previous dose-response studies^{1, 2, 3}, indicating that optimal catalytic activity and bioactivity can be reached at 1% of H₂O₂. Given the previous data, the inclusion of various permutations (varied Fer and SnF₂ concentrations) and to streamline our experiments/comparisons, we focused on 1% of H₂O₂. Lower concentrations of H₂O₂ may be effective in the combined treatment of Fer and SnF₂ because SnF₂ enhances the catalytic activity of Fer, which could be evaluated in future work. We have added the following information in the revised manuscript (page 18, last paragraph):

“Herein, we used 1% of H₂O₂ based on our previous dose-response studies^{18, 33, 43}. As SnF₂ enhances the catalytic activity of Fer, lower concentrations of H₂O₂ may also be effective in combination with Fer and SnF₂ for biofilm disruption and caries prevention.”

4. In line 192-193, it is mentioned that “consistent with dynamic light scattering (DLS) data, mixing Fer with Sn F2 did not seem to affect the size of Fer. However, there is no data for DLS in the manuscript.

Response: We apologize for the confusion. Actually, we have presented DLS data in the Supplementary Information (Supplementary Table 1). To make it clear, we have added the text “Supplementary Table 1” in the revised manuscript (page 10, first paragraph, in the revised manuscript):

“Consistent with dynamic light scattering (DLS) data (Supplementary Table 1), mixing Fer with SnF₂ did not affect the size of Fer.”

Reviewer #3 (Remarks to the Author):

The authors describe a study to examine the synergistic effect between the approved agents stannous fluoride and Ferumoxytol (aqueous iron oxide nanoparticle) to target biofilms associated with dental caries and to prevent enamel demineralisation. The results achieved are particularly significant within the field and compare very favourably with the established literature. This is a very well conducted study that examines the biological, chemical, and physical mechanisms involved. The work certainly supports the conclusions and claims made.

We thank the reviewer for the positive comments.

Attention should be paid to the following:

Intro / line 77 – reference should be made to salivary buffering capacity and how this can differ between individuals.

Response: We agree that saliva buffering capacity is an important point. Therefore, we have added this point in the revised manuscript (page 3, first paragraph):

“Additionally, current modalities, including high-dose fluoride treatments, are insufficient to prevent dental caries in high-risk individuals or individuals with a low salivary flow/buffering

capacity where pathogenic dental biofilms rapidly accumulate under sugar-rich diets and poor oral hygiene that enables firm bacterial adhesion to teeth.”

Results section needs to be carefully checked as regards correct use of English language.

Response: We carefully reviewed our manuscript for the correct use of English.

Results / line 335 – *S. mutans* – please state designation of strain – assume this is of human origin.

Response: We apologize for the confusion. We used *S. mutans* UA159 to infect rats and have added the required information in the revised manuscript (page 17, last paragraph).

“In this model, rat pups were infected with *S. mutans* UA159 (a cariogenic oral bacterium) and provided sucrose-containing food and water (Fig. 5a).”

Results / line 415 – beneficial species were increased. Further discussion need as regards properties e.g., nitrate-reducing capacity / interaction with pathogens.

Response: We thank the reviewer for this valuable suggestion. We have included additional discussion as follows on page 22:

“The treatment can affect the localized acidic microenvironment of plaque biofilm by modulating the growth of oral health-associated bacteria. *Rothia* is a nitrate-reducing oral bacteria that can generate nitrite in proximity to raise the local pH⁴⁴. In Fer+SnF₂ treatment, Sn-bound in the vicinity could serve as electron donors, facilitating nitrite and ammonia production by *Rothia*⁴⁵. Furthermore, this localized pH change may act as a triggering factor shifting relative abundance between *Streptococcus/Lactobacillus* (as acidogenic bacteria) and *Haemophilus*⁴⁶. Bacterial shifts of lactate-producing *Streptococcus/Lactobacillus* may also affect the abundance of lactate-utilizing bacteria, such as *Veillonella*, as observed in the treatment group.”

Results / line 432 – please comment on increases in richness / diversity. Only ‘main bacterial genera’ shown in fig. 6D – please comment on any significant changes at the species level. Were populations of *S. mutans* monitored in the mouse model?

Response: We did not find any statistical difference in richness/diversity. We have discussed it in the revised manuscript (page 21, last paragraph):

"All treatment groups showed no significant differences in alpha diversity between each group (Fig. 6a, b; $p > 0.05$, Willcox test)."

Regarding changes at species level, we were unable to accurately identify microbes to the species level using Illumina sequencing. We obtained sequences below 250 bp from the variable regions V1 to V3, while the complete 16S rRNA gene is 1550 bp. Therefore, partial coverage of the gene was analyzed for the microbiome. In addition, *Streptococcus* species have poor taxonomic resolution below the genus level⁴. Recently, it was reported other targets such as the 30S-S11 rRNA gene that could increase resolution and determine oral *Streptococci* profile with enhanced accuracy⁴. However, we used the universal bacterial gene 16S rRNA for analyzing the microbial community of the oral samples. Thus, *S. mutans* could not be identified and monitored using a partial region of the 16S rRNA gene. We have added this point as a limitation of our study (page 22, above first paragraph).

"In the current study, we could not monitor *S. mutans* using a partial region of the 16S rRNA gene. Moreover, *Streptococcus* species have poor taxonomic resolution below the genus level⁴⁷. Future studies using shotgun whole genome sequencing are warranted for monitoring functional microbiome changes at species-level with higher resolution and accuracy as well as microbial association network analyses to identify interspecies interactions."

References

1. Liu Y, *et al.* Topical ferumoxytol nanoparticles disrupt biofilms and prevent tooth decay in vivo via intrinsic catalytic activity. *Nature Communications* **9**, 2920 (2018).
2. Gao L, *et al.* Nanocatalysts promote *Streptococcus mutans* biofilm matrix degradation and enhance bacterial killing to suppress dental caries in vivo. *Biomaterials* **101**, 272-284 (2016).
3. Naha PC, *et al.* Dextran-coated iron oxide nanoparticles as biomimetic catalysts for localized and pH-activated biofilm disruption. *ACS Nano* **13**, 4960-4971 (2019).
4. O'Connell LM, Blouin T, Soule A, Burne RA, Nascimento MM, Richards VP. Optimization and evaluation of the 30S-S11 rRNA gene for taxonomic profiling of oral streptococci. *Applied and Environmental Microbiology* **88**, 00453-00422 (2022).

Reviewers' Comments:

Reviewer #1:

Remarks to the Author:

The authors made appropriate changes and answered all questions raised previously. I have no further comments.

Reviewer #2:

Remarks to the Author:

The authors have almost addressed all the concerns and questions of the reviewers and improved the quality of the manuscript. One more concern is that it is unclear if SnF₂ is still stable on Fer after catalysis. It has been known that SnF₂ also can react with H₂O₂. I am wondering if SnF₂ retains on Fer after H₂O₂ treatment. I suggest to characterize Sn on Fer after one or two cycle reaction with H₂O₂-TMB, either using the method for Figure 3c or ICP-MS.

Reviewer #3:

Remarks to the Author:

The authors have attended to the comments / issues raised by reviewer 3.

REVIEWER COMMENTS

Reviewer #1 (Remarks to the Author):

The authors made appropriate changes and answered all questions raised previously. I have no further comments.

Response: We thank the reviewer for helping to improve our manuscript and recommending it for publication.

Reviewer #2 (Remarks to the Author):

The authors have almost addressed all the concerns and questions of the reviewers and improved the quality of the manuscript. One more concern is that it is unclear if SnF₂ is still stable on Fer after catalysis. It has been known that SnF₂ also can react with H₂O₂. I am wondering if SnF₂ retains on Fer after H₂O₂ treatment. I suggest to characterize Sn on Fer after one or two cycle reaction with H₂O₂-TMB, either using the method for Figure 3c or ICP-MS.

Response: Thank you for your thoughtful comment. Your suggestion further improved the quality of our work, as we now have a better understanding of the stability of SnF₂ after catalysis. In response to your concern, we conducted additional experiments to investigate whether SnF₂ retains on Fer after H₂O₂ treatment.

Despite the known reactivity of SnF₂ with H₂O₂^{1, 2, 3}, our experimental findings indicate that a substantial portion of SnF₂ retains on Fer and is stable even after the catalytic reaction. We found that the stability of SnF₂ with Fer is comparable before and after catalytic reaction (Supplementary Fig. 10b). Notably, the addition of Fer increased the stability of SnF₂ by 10 times after H₂O₂ treatment compared with no Fer (Supplementary Fig. 11), whereas a slightly smaller amount of free tin ions is detected via ICP-OES after catalytic reaction (Supplementary Fig. 12). We have provided a detailed discussion of these findings in the revised manuscript (pages 16-17 and 33-34). Below, you will find the added information for your convenience:

“It has been reported that H_2O_2 can induce the oxidation of SnF_2 as H_2O_2 is a well-known oxidizing agent^{1, 2, 3}. To assess the stability of SnF_2 when mixed with Fer after the catalytic reaction, we conducted a series of experiments. Initially, we recorded UV-visible absorption spectra of SnF_2 in the presence and absence of H_2O_2 . As expected, the absorption curve of SnF_2 increased in the presence of H_2O_2 when compared to SnF_2 alone (Supplementary Fig. 10a), indicating that H_2O_2 caused the oxidation of SnF_2 . However, the absorption spectra of $\text{Fer}+\text{SnF}_2$ remained almost unchanged after the catalytic reaction (in the presence of H_2O_2), even after 60 min of catalysis (Supplementary Fig. 10b) as compared to Fer alone, suggesting that Fer may prevent the oxidation of Sn^{2+} . We also compared the absorbance of SnF_2 with or without the presence of Fer at 550 nm as a quantitative measurement of turbidity after 1 h incubation with H_2O_2 . As shown in Supplementary Fig. 11, the presence of Fer significantly reduced the oxidation of SnF_2 .

Supplementary Fig. 10. **a** Representative UV-visible absorption spectra of SnF_2 with or without H_2O_2 (10 min incubation) in 0.1 M sodium acetate buffer (pH 4.5). The addition of H_2O_2 resulted in a noticeable increase in the absorption spectrum of SnF_2 , indicating the oxidation of SnF_2 by H_2O_2 . **b** Representative UV-visible absorption spectra of Fer in combination with SnF_2 in the presence of H_2O_2 at different time points, as indicated, in 0.1 M sodium acetate buffer (pH 4.5). The absorption spectra of $\text{Fer}+\text{SnF}_2$, upon incubation with H_2O_2 at various time points, do not change appreciably when compared to Fer alone, even after 60 min of catalysis, suggesting Fer may prevent oxidation of SnF_2 .

Supplementary Fig. 11. Comparison of the absorbance of SnF₂ (0.1 mg/ml) at 550 nm in the presence of H₂O₂ (0.1%, v/v) with or without Fer (0.1 mg of Fe/ml) in 0.1 M sodium acetate buffer (pH 4.5). The absorbance of Fer was used as the background for the Fer+SnF₂+H₂O₂ group. The absorbance measurements were taken after 1 h incubation in the presence of H₂O₂. The data are presented as mean ± standard deviation (n=3). ****p* < 0.001; one-way ANOVA followed by Tukey test.

To further evaluate the stability of SnF₂ when mixed with Fer after H₂O₂ treatment, we measured the concentration of free tin ions with or without H₂O₂ exposure by ICP-OES (Supplementary Fig. 12). We found that the amount of free tin ions was slightly less after the catalytic reaction (i.e., H₂O₂ treatment), thereby indicating minimal loss of SnF₂ stabilization. Based on these experiments, it is apparent that the majority of SnF₂ remains bound on Fer and stable even after the catalytic reaction. These findings provide further evidence that Fer acts as a stabilizing agent for SnF₂, effectively reducing its oxidation in the presence of H₂O₂.

Supplementary Fig. 12. Comparison of the concentration of free tin ions in the filtrate when SnF₂ (1 mg/ml) was mixed with Fer (1 mg of Fe/ml) in 0.1 M sodium acetate buffer (pH 4.5) after 10 min incubation in the absence and presence of H₂O₂ (1%, v/v), as determined by ICP-OES. The data are presented as mean ± standard deviation (n=3). **p* < 0.05; one-way ANOVA followed by Tukey test.

Method:

Stability study of SnF₂ after catalytic reaction

To investigate the extent of SnF₂ oxidation after catalytic reaction, SnF₂ (1 mg/ml) was mixed with Fer (1 mg of Fe/ml) in 0.1 M sodium acetate buffer (pH 4.5), and then H₂O₂ (1 %, v/v) was added to the solution to initiate the reaction. After incubating the mixture for the predetermined time with H₂O₂, the absorption spectra of the solutions were recorded following a 10-fold dilution. Furthermore, absorption spectra of SnF₂ (1 mg/ml) were recorded in the presence and absence of H₂O₂ (1%, v/v) (10 min incubation in the presence of H₂O₂) in 0.1 M sodium acetate buffer (pH 4.5). Additionally, the absorption spectrum of the diluted SnF₂ solution was measured after 60 min incubation in the presence of H₂O₂.

Determination of free tin ions after the catalytic reaction

The free tin ions from the combination of Fer+SnF₂, in the presence and absence of H₂O₂, was investigated using ICP-OES. Briefly, SnF₂ (1 mg/ml) was mixed with Fer (1 mg of Fe/ml) in 0.1 M sodium acetate buffer (pH 4.5) and then incubated for 24 h at room temperature. The solution was then further incubated for 10 min with or without H₂O₂ (1%, v/v) to initiate the catalytic reaction. Afterward, free tin ions were collected from the filtrate by centrifugation (1 h; 4000 rpm) using ultrafiltration tubes (3 kDa MWCO). Subsequently, the filtrate was digested in nitric acid and diluted with DI water before analysis by ICP-OES.”

Reviewer #3 (Remarks to the Author):

The authors have attended to the comments / issues raised by reviewer 3.

Response: We thank the reviewer for helping to improve our manuscript and recommending it for publication.

References

1. Hefferren JJ. Qualitative and quantitative tests for stannous fluoride. *Journal of Pharmaceutical Sciences* **52**, 1090-1096 (1963).
2. Desmau M, Alsina MA, Gaillard J-F. XAS study of Sn speciation in toothpaste. *Journal of Analytical Atomic Spectrometry* **36**, 407-415 (2021).
3. Denes G, Laou E, Muntasar A. Reaction of stannous fluoride in hydrogen peroxide. *Hyperfine Interactions* **90**, 429-433 (1994).

Reviewers' Comments:

Reviewer #2:

Remarks to the Author:

The authors have confirmed the stability of SnF₂ on Fer and after catalysis with H₂O₂. Such stability property is beneficial for Fer+SnF₂ to treat dental caries. The concerns of the reviewer have been addressed. Thus, the current version of the manuscript can be accepted for publication.